# PIEZO1 and PECAM1 interact at cell-cell junctions and partner in endothelial force sensing

Eulashini Chuntharpursat-Bon [1,7✉], Oleksandr V. Povstyan[1,6], Melanie J. Ludlow[1,6], David J. Carrier[1,2], Marjolaine Debant[1], Jian Shi[1], Hannah J. Gaunt[1], Claudia C. Bauer[1], Alistair Curd [3], T. Simon Futers [1], Paul D. Baxter[1], Michelle Peckham [3,4], Stephen P. Muench [2,4], Antony Adamson [5], Neil Humphreys[5], Sarka Tumova [1], Robin S. Bon [1,4], Richard Cubbon[1], Laeticia Lichtenstein [1] & David J. Beech [1,7✉]

Two prominent concepts for the sensing of shear stress by endothelium are the PIEZO1 channel as a mediator of mechanically activated calcium ion entry and the PECAM1 cell adhesion molecule as the apex of a triad with CDH5 and VGFR2. Here, we investigated if there is a relationship. By inserting a non-disruptive tag in native PIEZO1 of mice, we reveal in situ overlap of PIEZO1 with PECAM1. Through reconstitution and high resolution microscopy studies we show that PECAM1 interacts with PIEZO1 and directs it to cell-cell junctions. PECAM1 extracellular N-terminus is critical in this, but a C-terminal intracellular domain linked to shear stress also contributes. CDH5 similarly drives PIEZO1 to junctions but unlike PECAM1 its interaction with PIEZO1 is dynamic, increasing with shear stress. PIEZO1 does not interact with VGFR2. PIEZO1 is required in $Ca^{2+}$-dependent formation of adherens junctions and associated cytoskeleton, consistent with it conferring force-dependent $Ca^{2+}$ entry for junctional remodelling. The data suggest a pool of PIEZO1 at cell junctions, the coming together of PIEZO1 and PECAM1 mechanisms and intimate cooperation of PIEZO1 and adhesion molecules in tailoring junctional structure to mechanical requirement.

[1] School of Medicine, University of Leeds, Leeds LS2 9JT, UK. [2] School of Biomedical Sciences, University of Leeds, Leeds LS2 9JT, UK. [3] School of Molecular and Cellular Biology, University of Leeds, Leeds LS2 9JT, UK. [4] Astbury Centre for Structural Molecular Biology, University of Leeds, Leeds LS2 9JT, UK. [5] Faculty of Biology, Medicine and Health, University of Manchester, AV Hill Building, Manchester M13 9PT, UK. [6] These authors contributed equally: Oleksandr V. Povstyan, Melanie J. Ludlow. [7] These authors jointly supervised this work: Eulashini Chuntharpursat-Bon, David J. Beech. ✉email: medechu@leeds.ac.uk; d.j.beech@leeds.ac.uk

The endothelium comprises a monolayer of endothelial cells at the inner surface of all arteries, veins, capillaries and lymphatics. A key function is to provide a selective barrier to the exchange of substances and cells between blood and tissue[1,2]. Endothelial cell permeability, transmembrane proteins and the structures between endothelial cells such as adherens junctions all contribute to the barrier[1,2]. These systems operate in the context of mechanical forces caused by the heartbeat, skeletal and smooth muscle-induced movements, gravity, interstitial pressure, cell-cell interactions and cell-matrix interactions[3–7]. The flow of blood and lymph have particular importance because of the shear stress they generate at the endothelial surface[3,4,8]. Shear stress varies in ferocity and orientation depending on vascular architecture and must be coordinated with other forces such as pulsatile circumferential strain[3]. There has been progress in understanding how these forces interact with the biological mechanisms but much remains opaque; especially regarding the sensing of local mechanical forces, the integration of such sensing with other endothelial mechanisms and the subcellular organisation of the sensing mechanisms.

PECAM1 (Platelet Endothelial Cell Adhesion Molecule 1 or CD31) is a candidate mediator of endothelial responses to shear stress[9], even though its predominant localisation at adherens junctions hides most of it from shear stress[1,4,8]. CDH5 (Cadherin-5 or Vascular endothelial (VE) cadherin) is included in the PECAM1 hypothesis along with vascular endothelial growth factor receptor 2 (VGFR2), leading to the concept of a PECAM1 triad as the shear stress sensor[10–16]. PECAM1 and CDH5 are key endothelial cell adhesion molecules. They are single-pass membrane proteins that mediate cell contact and junctional integrity[2,17–21]. PECAM1 belongs to type-I membrane glycoprotein and immunoglobulin super-families[17,20,22]. It has an extracellular region containing immunoglobulin-like domains, a transmembrane region of one α-helix and a cytoplasmic region containing tyrosine regulatory motifs. It is often used to identify endothelial cells, though is also expressed and functional in leucocytes and platelets[17,20]. CDH5 belongs to a family of transmembrane $Ca^{2+}$-dependent adhesion molecules[19,21,23]. Like PECAM1, it has an extracellular domain that mediates homotypic interactions and intracellular regulatory sites, which, in this case, bind β- or γ-catenin to promote cytoskeletal interaction[19,21]. CDH5 is likened to a biochemical Velcro and may also be a mechanical transducer[24].

PIEZO1 protein was recognised later[25–28] and suggested as a mediator of endothelial responses to shear stress[29–32]. It assembles as trimers to form large $Ca^{2+}$-permeable non-selective cationic channels, each with 114 (3×38) membrane-spanning segments[25,27,33–35]. These channels are exquisitely sensitive to activation by various mechanical forces[25–29]. Importantly in the context of endothelial cell and other cell biology, shear stress activates PIEZO1 channels[29–32,36–38]. PIEZO1 is strongly, although not uniquely, expressed in endothelial cells[26,29]. It is required for endothelial cell responses to force such as their alignment to the direction of fluid flow and the activation of endothelial nitric oxide synthase (NOS3)[29,30,32]. Shear stress-activated PIEZO1 channels are present in the apical endothelial membrane[38,39] and so they are ideally located to mediate the sensing of shear stress, albeit potentially via or amplified by intermediates of the glycocalyx[40] and spectrin cytoskeleton[41].

The study described here was motivated by a desire to understand the apparently competing ideas for sensing of shear stress by the PECAM1 triad and PIEZO1, but this relationship is also important to investigate because of the relationship between PIEZO1 and endothelial cell-cell junctions, and thus the integrity of the endothelial cell monolayer and its permeability. Endothelial-specific PIEZO1 disruption in mice suppresses or enhances vascular permeability caused by excess vascular or alveolar pressures[42,43]. In mice, endothelial PIEZO1 is required for leucocyte diapedesis[44], which is the process of leucocytes passing through the endothelial monolayer. In cultured mouse or human endothelial cells, stimulation of PIEZO1 by a small-molecule agonist (Yoda1) straightens CDH5 junctions[45] and PIEZO1 depletion inhibits stretch-evoked remodelling of endothelial cell adherens junctions[46]. These data suggest that PIEZO1 is an important regulator of cell-cell junctions in addition to having a role in sensing shear stress. In mechanistic interpretations of these and other such data, PIEZO1 is currently placed at the apical endothelial surface, signalling from a distance to cell-cell junctions and other mechanisms[1,47]. We agree with such a location of PIEZO1 channels[38,39] but hypothesise greater complexity.

Here we suggest a pool of PIEZO1 at adherens junctions, interactions of PIEZO1 with PECAM1 and CDH5 and roles of PIEZO1 in cell junction remodelling. First, to enable specific labelling of endogenous PIEZO1 and thus determination of its localisation in vivo, we engineered a mouse with a non-disruptive tag in PIEZO1.

## Results

### Genetic engineering of mice enables insertion of a non-disruptive HA tag in native PIEZO1.

For definitive localisation of endogenous PIEZO1, we genetically modified mice to encode a non-disruptive haemagglutinin (HA) tag in the C-terminal Extracellular Domain (CED) of native PIEZO1 (Supplementary Fig. 1). Activity of these PIEZO1$^{HA}$ channels was recorded in endothelium of mesenteric artery where we previously showed the presence of wild-type PIEZO1 (PIEZO1$^{WT}$) channels activated by fluid flow, membrane stretch or the PIEZO1 small-molecule agonist Yoda1[38,39]. The PIEZO1$^{HA}$ channels studied in excised outside-out endothelial membrane patches are similar to PIEZO1$^{WT}$ channels in their activation by fluid flow, unitary conductance and sensitivity to inhibition by gadolinium ions ($Gd^{3+}$), which non-specifically inhibit PIEZO1 channels[25,38] (Fig. 1a–c cf Fig. 1f–h, k). There is basal activity of the PIEZO1$^{HA}$ channels in the static (no-flow) condition (Fig. 1a, k) as reported previously and shown independently here for PIEZO1$^{WT}$ channels[38,39] (Fig. 1f, k). Membrane potential recordings from multicellular endothelial fragments similarly obtained freshly from the arteries of PIEZO1$^{HA}$ mice depolarise in response to fluid flow (Fig. 1d, e, k), again similar to wild-type endothelium (Fig. 1i–k) and as expected for PIEZO1 activity[38,39]. Consistent with similar properties of native PIEZO1$^{HA}$ and PIEZO1$^{WT}$ channels, PIEZO1$^{HA}$ mice appear healthy and breed normally (Supplementary Fig. 2a, b), in contrast to PIEZO1 knockouts, which are embryonic lethal[30,37]. Red blood cells (RBCs) are an abundant and readily purified cell type that expresses PIEZO1[48,49] and so they were used for PIEZO1$^{HA}$ detection by western blotting, which reveals PIEZO1$^{HA}$ protein of the expected mass (Supplementary Fig. 2c), again similar to that of PIEZO1$^{WT}$[30]. The data suggest suitability of PIEZO1$^{HA}$ mice for determining native localisation of PIEZO1.

### PIEZO1$^{HA}$ expression pattern is similar to that of PECAM1 in endothelium.

To explore PIEZO1 localisation, we studied the retina where the entire vascular tree is imaged in one sample (Supplementary Fig. 3). PIEZO1$^{HA}$ is detected in retinal veins (Fig. 2a, b cf further data and controls shown in Supplementary Fig. 4a–e). Close inspection shows PIEZO1 at areas of cell-cell contact where PECAM1 is predominantly expressed (+PECAM1) (Fig. 2a, c). In non-junctional areas without PECAM1 (−PECAM1), the normalised fluorescence signal is

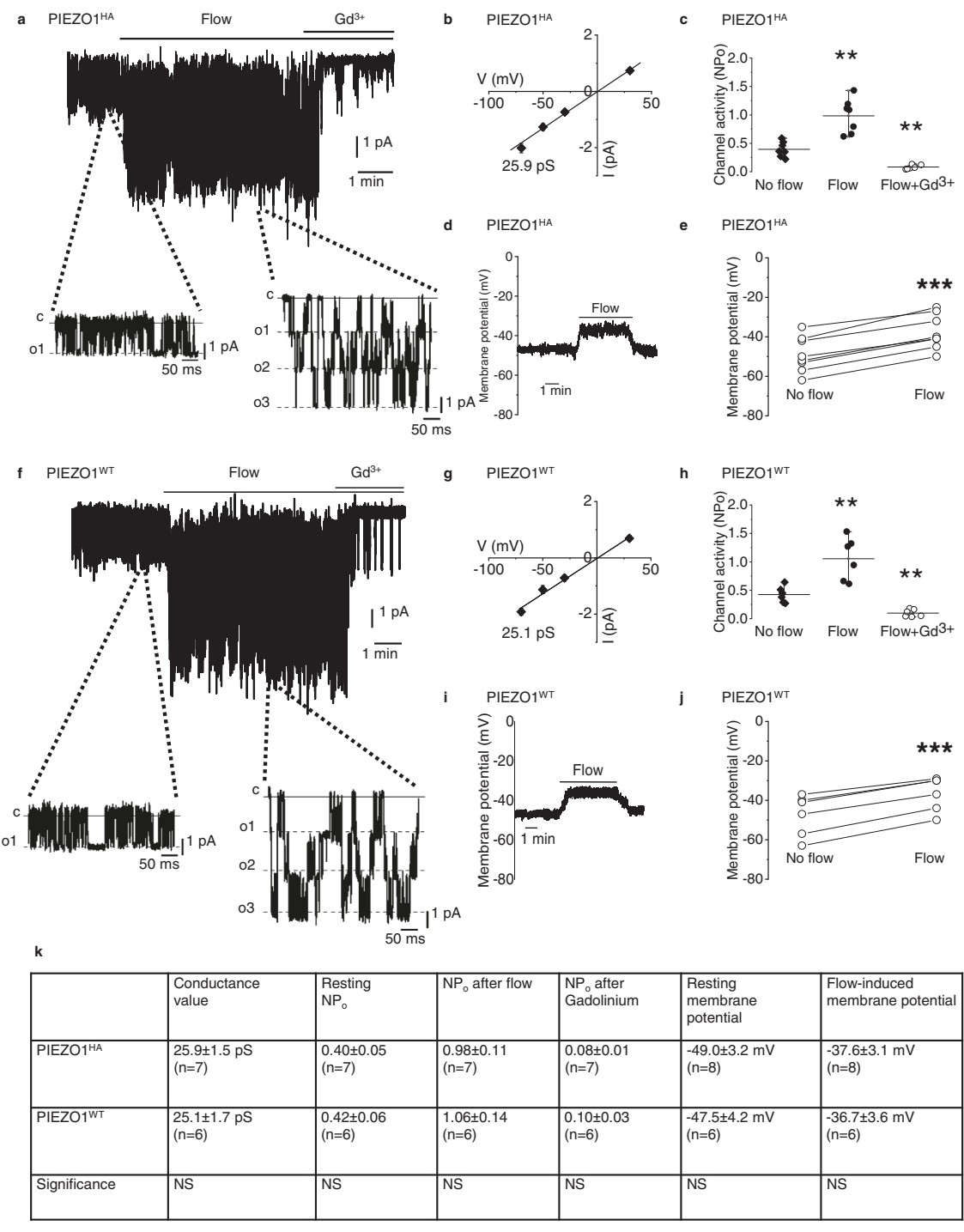

| | Conductance value | Resting $NP_o$ | $NP_o$ after flow | $NP_o$ after Gadolinium | Resting membrane potential | Flow-induced membrane potential |
|---|---|---|---|---|---|---|
| PIEZO1HA | 25.9±1.5 pS (n=7) | 0.40±0.05 (n=7) | 0.98±0.11 (n=7) | 0.08±0.01 (n=7) | -49.0±3.2 mV (n=8) | -37.6±3.1 mV (n=8) |
| PIEZO1WT | 25.1±1.7 pS (n=6) | 0.42±0.06 (n=6) | 1.06±0.14 (n=6) | 0.10±0.03 (n=6) | -47.5±4.2 mV (n=6) | -36.7±3.6 mV (n=6) |
| Significance | NS | NS | NS | NS | NS | NS |

close to 0.4 (Fig. 2c), which is at or near the background values obtained from PIEZO1WT tissues under similar conditions and the same microscope settings (Supplementary Fig. 4a–c, f–k). PIEZO1HA is not detected in retinal artery, although PIEZO1 may be in these arteries[50] but below the threshold for detection in our assay (Supplementary Fig. 4f–k). PIEZO1HA is detected in retinal intermediate and deep capillaries (Supplementary Fig. 4l–q). The data suggest that PIEZO1 is at points of endothelial cell-cell contact with PECAM1.

**Pharmacological activation of PIEZO1 disrupts PECAM1 structural organisation**. To determine if PIEZO1 has

functional implications for in situ PECAM1, we infused PIEZO1 channel small-molecule agonist Yoda1[51] for 30 min in vivo, using exsanguinated mice to minimise problems due to potential Yoda1 instability and plasma protein binding. Mice were then perfusion-fixed and retinal vasculature was stained. We showed previously that the effects of Yoda1 on mouse endothelial cells are abolished by endothelial-specific PIEZO1 deletion[38]. In the retina, Yoda1 disorders the pattern of PECAM1, notably in vein (Fig. 2d, e, Supplementary Fig. 5a) but not artery (Supplementary Fig. 5b–d). The venous specificity aligns with the venous localisation (Fig. 2a, c), consistent with the idea that Yoda1 acts via PIEZO1. The data suggest that PIEZO1 is capable of regulating the organisation of PECAM1 at endothelial cell-cell junctions.

**Fig. 1 Genetically engineered PIEZO1$^{HA}$ has comparable activity to wild-type PIEZO1 (PIEZO1$^{WT}$) in mice. a** Example current recording from an outside-out patch from freshly isolated endothelium of second-order mesenteric artery of PIEZO1$^{HA}$ mouse. Holding potential, −70 mV. Fluid flow, 20 µl.s$^{-1}$. Gadolinium (III) ion (Gd$^{3+}$), 10 µM. Currents on expanded time-base are shown below in which c indicates the closed channel current level and o1, o2 and o3 the open channel current levels for simultaneous openings of up to 3 channels. **b** Mean ± s.e.mean unitary current amplitudes for flow-induced ion channel activity as in **a** plotted against holding voltage ($n = 7$ independent recordings). The fitted line indicates unitary conductance of 25.9 pS. **c** Channel activity indicated by NP$_o$ (number of channels per patch × probability of channel opening) for experiments of the type exemplified in **a** for no flow and flow conditions with or without Gd$^{3+}$. Individual data points for each experiment are represented by symbols, superimposed on which are the mean ± s.e.mean values (**$P_{Flow}$ = 0.0001703, **$P_{Flow+Gd3+}$ = 0.000165, $n = 7$ for each group). **d** Example trace of membrane potential measured from freshly isolated endothelium of second-order mouse mesenteric artery of PIEZO1$^{HA}$ mouse. Fluid flow, 20 µl.s$^{-1}$. **e** Individual data points representing membrane potentials before and after flow as exemplified in **d** (***$P$ = 0.00000343) ($n = 7$). Data points connected by a line were from the same recording. **f** Example current recording from an outside-out patch from freshly isolated endothelium of second-order mesenteric artery of PIEZO1$^{WT}$ mouse. Holding potential, −70 mV. Fluid flow, 20 µl.s$^{-1}$. Gadolinium (III) ion (Gd$^{3+}$), 10 µM. **g** Mean ± s.e.mean unitary current amplitudes for flow-induced ion channel activity as in **b** plotted against holding voltage ($n = 6$ independent recordings). The fitted line indicates unitary conductance of 25.9 pS. **h** Channel activity indicated by NP$_o$ (number of channels per patch × probability of channel opening) for experiments of the type exemplified in **f** for no flow and flow conditions with or without Gd$^{3+}$. Individual data points for each experiment are represented by symbols, superimposed on which are the mean ± s.e.mean values (**$P_{Flow}$ = 0.000141, **$P_{Flow+Gd3+}$ = 0.000241, $n = 6$ for each group). **i** Example trace of membrane potential measured from freshly isolated endothelium of second-order mouse mesenteric artery of PIEZO1$^{WT}$ mouse. Fluid flow, 20 µl.s$^{-1}$. **j** Individual data points representing membrane potentials before and after flow as exemplified in **i** ($P$ = 0.0000375, $n = 6$). Data points connected by a line were from the same recording. **k** Table of data comparing values from PIEZO1$^{HA}$ mouse and PIEZO1$^{WT}$ mouse, (mean ± s.e.m.). NS indicates no statistically significant difference between PIEZO1$^{HA}$ and PIEZO1$^{WT}$. For the outside-out patch and membrane potential recordings, the external solution consisted of: 135 mM NaCl, 4 mM KCl, 2 mM CaCl$_2$, 1 mM MgCl$_2$, 10 mM glucose and 10 mM HEPES (titrated to pH 7.4 with NaOH). The patch pipette contained: 145 mM KCl, 1 mM MgCl$_2$, 0.5 mM EGTA and 10 mM HEPES (titrated to pH 7.2 with KOH).

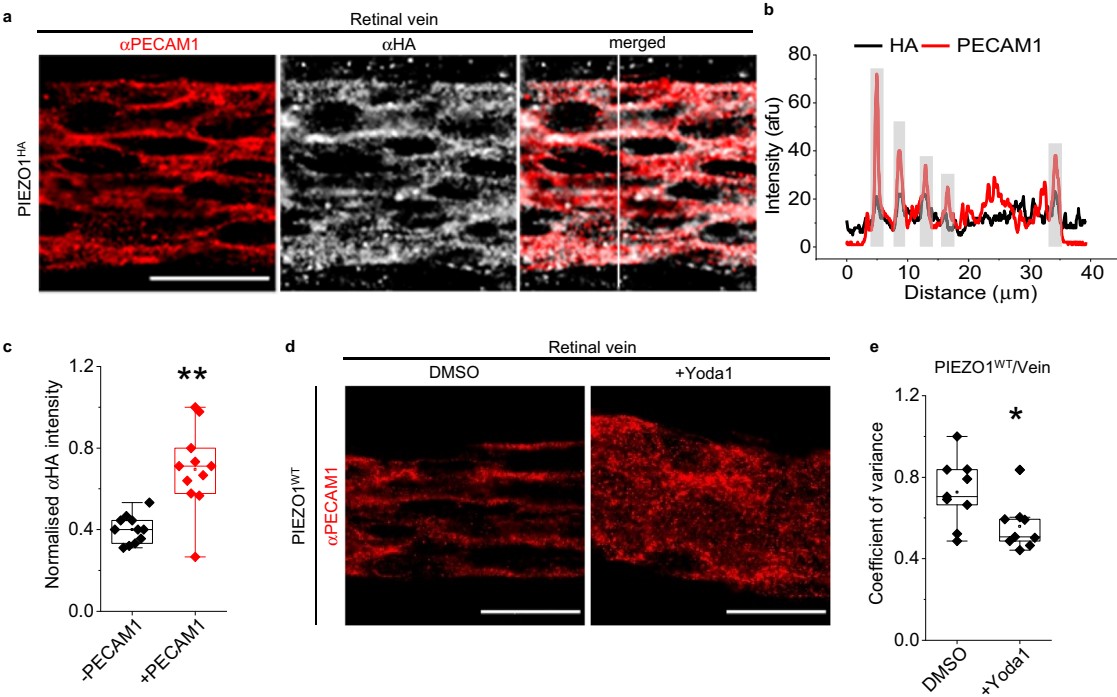

**Fig. 2 Endogenous PIEZO1 is close to and can disrupt PECAM1 at cell-cell junctions. a** Images for retinal vein of PIEZO1$^{HA}$ mouse after immuno-staining with αPECAM1 (red) and αHA (grey). The image on the right side is a merged αPECAM1 and αHA image. Scale bar, 20 µm. **b** Line-intensity plot for the vertical scan line superimposed in the merged image of **a**. Grey highlighting indicates cell-cell junctions. **c** Box-plot quantification of image intensity for PIEZO1$^{HA}$ retinal veins stained with αHA antibody, shown in arbitrary fluorescence units (afu) normalised to the background measurements in the same afu. **$P$ = 0.00127 for the comparison of the regions with (+) or without (−) PECAM1. Superimposed data points are average intensity of individual images ($N = 11$). Data are for $n = 3$ independent experiments. **d** Data obtained after WT mice were infused for 30 min with standard bath solution (SBS) containing DMSO (the solvent for Yoda1) or 3 µM Yoda1. Representative images of **d** retinal vein immuno-stained with αPECAM1 antibody. Scale bar, 20 µm. Box-plots **e** show coefficients of variance calculated from scan lines that were vertical to a blood vessel oriented from left to right (*$P$ = 0.02728 for Yoda1 *cf* DMSO in vein). Lower variance indicates less organised structure. $n = 3$ independent experiments and 3 replicates were used in each case. Superimposed data points are the coefficient of variance for individual images (DMSO, $N = 9$ and +Yoda1, $N = 9$).

**PIEZO1 channel function decreases when the abundance of PECAM1 increases.** We considered the possibility of a two-way relationship between PIEZO1 and PECAM1, with PECAM1 affecting PIEZO1 activity. We studied human umbilical vein endothelial cells (HUVECs) in culture as a model endothelial cell system for mechanistic studies. As HUVECs increase in density (Fig. 3a–c), they express more PECAM1 (Fig. 3d, e) and there is more PECAM1 at cell-cell junctions (Fig. 3b). There is

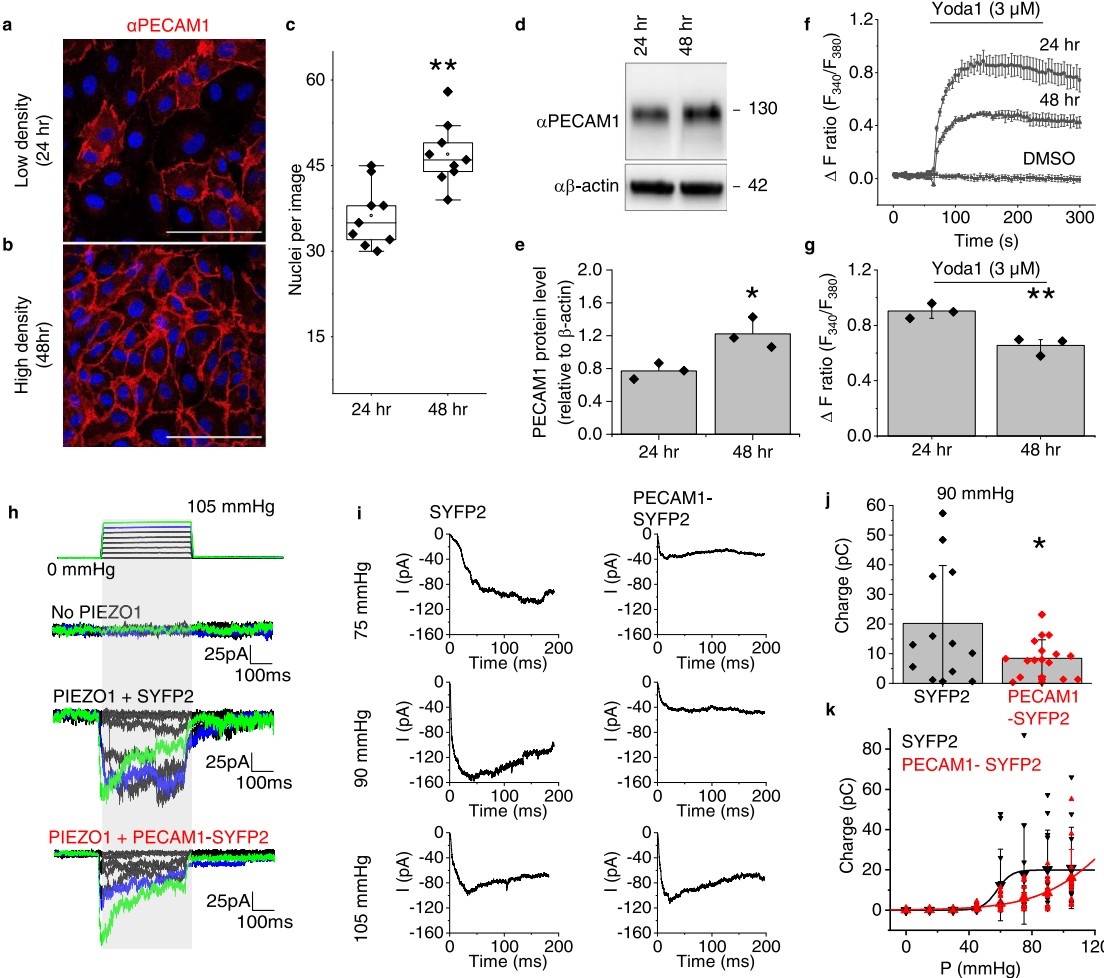

**Fig. 3 PECAM1 suppresses PIEZO1 function. a–g** HUVEC data. Representative of 3 independent experiments, confocal images at low **a** and high **b** cell density, 24 and 48 h after plating. Cells were stained with αPECAM1 (red) and DAPI (nuclei, blue). Scale bar, 100 μm. **c** For experiments of the type in **a** and **b**, box plots of the number of nuclei per field of view at the 24 and 48 h time-points ($n = 3$, $**P = 0.00264$). Superimposed data points are nuclei numbers for individual images (24 h, $N = 9$ and 4 h, $N = 9$). **d** Representative western blot and **e** quantification of the PECAM1 in such blots **d** normalised to β-actin (mean ± s.e. mean, $n = 3$, $*P = 0.04216$, two sample t-test). Superimposed data points are quantification of independent western blots. **f, g** As for **a** and **b** but fura2-based $Ca^{2+}$ measurement data. **f** Representative time-series traces showing effects of 3 μM Yoda1 at the two cell-culture time-points, compared with vehicle control DMSO (3 replicates per data point). **g** For data of the type in **f**, mean ± s.e.mean ($n = 3$) for peak $Ca^{2+}$ signals evoked by Yoda1 ($**P = 0.00705$, two sample t-test). Superimposed data points are the mean intensity ratios for each independent repeat. **h–k** Data from outside-out patch recordings on T-Rex™-293 cells. **h** Example original current traces for empty cells without incorporation of PIEZO1 (No PIEZO1) or T-REx-293 cells with tetracycline-induced expression of human PIEZO1 after transient transfection with SYFP2 (PIEZO1 + SYFP2) or human PECAM1-SYFP2 (PIEZO1 + PECAM1-SYFP2). The holding potential was −80 mV and 200-ms positive pressure pulses were applied from 0 to 105 mmHg in 15 mmHg increments at intervals of 12 s as illustrated above the current traces (traces for 90 and 105 mmHg are highlighted in blue and green respectively). **i** Average PIEZO1 currents for cells transfected with SYFP2 ($n = 12$) or PECAM1-SYFP2 ($n = 13$) for each of the indicated pressure steps. **j** For the same type of data as **h**, electric charge compared at 90 mmHg for the two groups, suggesting statistically significant difference by t-test ($*P = 0.01797$). Superimposed individual data points for PIEZO1 + SYFP2 (black, $n = 12$) and PIEZO1 + PECAM1-SYFP2 (red, $n = 15$). **k** For the same type of data as **h**, mean ± S.D. and superimposed individual data points for PIEZO1 + SYFP2 (black, $n = 11$) and PIEZO1 + PECAM1-SYFP2 (red, $n = 15$), showing integrated electrical charge per pressure step plotted against pressure. F-test indicated statistically significant different between the pressure curves of the two conditions ($***P = 7.45 \times 10^{-8}$).

simultaneously reduced PIEZO1-mediated $Ca^{2+}$ entry (Fig. 3f, g). The data suggest that increased abundance of PECAM1 reduces PIEZO1 channel function.

**PECAM1 inhibits PIEZO1 mechanical sensitivity**. To more directly test if PECAM1 inhibits PIEZO1 activity, we made patch-clamp recordings of PIEZO1 channel activity in modified HEK 293 cells engineered for conditional overexpression of human PIEZO1 (hPIEZO1 T-Rex™-293 cells) and transfected with PECAM1 tagged with a fluorescent marker protein Super Yellow Fluorescent Protein 2 (SYFP2) or the control, SYFP2 only. We used excised outside-out

patches for data collection from the surface membrane and applied positive pressure steps to mechanically activate the channels, as previously reported[52]. As expected[52], cells expressing PIEZO1 show prominent mechanically activated macroscopic currents (Fig. 3h, Supplementary Fig. 6). PECAM1 inhibits the PIEZO1 activity quantified as the total charge flow per pressure step up to 90 mmHg (Fig. 3i, j), shifting the pressure-response curve to the right (Fig. 3k). Inactivation is unaffected (Supplementary Fig. 6). Mouse PIEZO1 is similarly modulated (Supplementary Fig. 7). The data suggest that PECAM1 reduces the mechanical sensitivity of PIEZO1, consistent with the idea of a two-way relationship.

**PECAM1 interacts with PIEZO1 N-terminus.** N-terminal regions of PIEZO1 determine mechanical sensitivity[33,35,53] and so PECAM1 may interact with these regions. We tested if there is interaction between PIEZO1 and PECAM1 by coexpressing PECAM1 with Halo-tagged human PIEZO1 or Halo tag alone as a control. Using an antibody to the Halo Tag, PECAM1 coprecipitates with Halo-tagged PIEZO1, but not with the Halo tag alone, suggesting that PECAM1 and PIEZO1 interact (Supplementary Fig. 8a). Proteins were also purified using GST-anti-GFP nanobodies[54] from HEK 293 cells coexpressing PECAM1 and green fluorescent protein (GFP)-tagged PIEZO1 (PIEZO1-GFP) and then purified by size-exclusion chromatography. There is copurification of PECAM1 with PIEZO1, along with the detached GST-anti-GFP nanobody, again suggesting interaction (Supplementary Fig. 8b). We next generated HA-tagged PIEZO1 deletion constructs that retained the N-terminus (Supplementary Fig. 8c). We did not delete the N-terminus because of its importance for surface trafficking[55]. The constructs were coexpressed with PECAM1 and precipitated using anti-HA antibody. PIEZO1 without HA was a control for non-specificity and did not precipitate but all HA-tagged PIEZO1 constructs precipitated, including the shortest N-terminal fragment T6 (Supplementary Fig. 8d). The data suggest that PECAM1 inhibits PIEZO1 by interacting with its N-terminal regions.

**Reconstituted PIEZO1 and PECAM1 are at cell-cell junctions and physically close.** To investigate molecular details of PECAM1-PIEZO1 relationships in subcellular regions, we sought a host cell reconstitution system for high resolution light microscopy studies. African green monkey kidney COS-7 cells are such cells. They normally express little or no PIEZO1 or PECAM1, conferring a relatively null background. Exogenous PECAM1 was previously shown to naturally accumulate at COS-7 cell-cell junctions through diffusion trapping, suggesting suitability of these cells as a PECAM1 host[56]. The cells allow reconstitution of the PECAM1 triad and endothelial cell-like alignment to shear stress[11]. We therefore transfected human PECAM1 into COS-7 cells and labelled it with antibody targeted to PECAM1 extracellular N-terminus. HA was engineered into the human PIEZO1 C-terminal extracellular domain (CED) and transfected into COS-7 cells for specific labelling with the antibody targeted to HA. Cells were unpermeabilised, thereby allowing selective labelling of surface membrane proteins, and grown to confluence, so that they had cell-cell junctions. Transfection efficiency was optimised to minimise protein abundance while still enabling detection, resulting in only some cells being transfected and visualised.

Stimulated emission depletion (STED) microscopy was used for imaging at ~50 nm spatial resolution (Supplementary Fig. 9a–d). PECAM1 is located in puncta concentrated at cell junctions (Supplementary Fig. 9b). PIEZO1[HA] puncta are slightly larger (Supplementary Fig. 9c). Merged PECAM1 and PIEZO1[HA] images show that the two puncta are close to each other, particularly at cell-cell junctions (Supplementary Fig. 9d). We modelled the distribution of distances between PIEZO1[HA] and PECAM1 with non-Gaussian distributions as described previously[57,58] (see also the Methods section). From the fitted distribution of distances between PIEZO1[HA] particles and their nearest PECAM1 particles, we found frequent proximity averaging 34 nm and occasional proximity averaging 169 nm at cell-cell junctions (Supplementary Fig. 9e–h). At non-junctional regions, models fitted poorly, resulting in parameter uncertainties greater than parameter estimates, but proximity of ~50 nm is inferred from inspection of the histogram (Supplementary Fig. 9i–k). The data suggest close proximity of the reconstituted

proteins at cell-cell junctions, consistent with the two proteins interacting.

**PECAM1 drives PIEZO1 to cell-cell junctions.** To investigate the PIEZO1-PECAM1 relationship in more detail, fluorescence lifetime imaging microscopy (FLIM) was used for quantification of Förster Resonance Energy Transfer (FRET), which occurs at distances of less than 10 nm. On PIEZO1 we engineered a donor fluorophore (mTurquoise2) and on PECAM1 an acceptor fluorophore, for which we used SYFP2. We inserted a linker between the target protein and fluorescent tag for both constructs. Expressed alone, PIEZO1-mTurquoise2 is primarily around nuclei (N) and in endoplasmic reticulum (ER) but not at cell-cell junctions (Fig. 4a). By contrast, when coexpressed with PECAM1-SYFP2, the PIEZO1 enriches at points of cell-cell contact (Fig. 4b). The data suggest that PECAM1 drives a pool of PIEZO1 to cell-cell junctions.

**Detected PIEZO1-PECAM1 proximity is similar to that of PIEZO1-PIEZO1.** The fluorescence lifetime of PIEZO1-mTurquoise2 (expressed alone) peaks at ~4 ns (Fig. 4a, c). When coexpressed with PECAM1-SYFP2, there is shortening to ~3.5 ns, suggesting that the two fluorophores are within 10 nm (Fig. 4b, c), which is a distance lower than the width of one PIEZO1 channel (~20 nm)[33]. Comparable lifetime shortening occurs when PIEZO1-mTurquoise2 is co-expressed with PIEZO1-SYFP2 (Supplementary Fig. 10). PIEZO1 forms trimeric ion channels in which 3 PIEZO1s are physically bound[27,33]. Because of the comparable FRET of PIEZO1-PECAM1 and PIEZO1-PIEZO1, we suggest that PIEZO1 and PECAM1 interact as closely as two PIEZO1s.

**Mutation of C-terminal residues in PECAM1 prevents or reduces PIEZO1 interaction.** Because the FRET signal in the above PIEZO1-PECAM1 studies originated intracellularly (Fig. 4a–c), we hypothesised importance of intracellular (C-terminal) regions of PECAM1. To test this hypothesis, 5 amino acid residues in the C-terminus of PECAM1 were mutated based on prior knowledge of PECAM1 (Supplementary Fig. 11). Wild-type PECAM1-SYFP2 and mutant PECAM1-SYFP2 were then co-expressed with PIEZO1-mTurquoise2 in COS-7 cells and lifetimes determined at non-junctional regions and junctions identified by accumulated PECAM1 (Fig. 4d, e, Supplementary Fig. 12a–e). The Y713F mutation prevents FRET at both locations and is less able to drive PIEZO1 to junctions (Fig. 4d, e, Supplementary Fig. 12f, g), while retaining normal abundance and localisation (Fig. 4g *cf* 4f). FRET occurs with the other 4 mutants but is reduced for C622A and S700F, particularly in non-junctional regions (Fig. 4d). There is normal abundance and localisation (shown for C622A in Fig. 4h). The data suggest that C-terminal structure of PECAM1 and particularly Y713 influence the interaction of PECAM1 with PIEZO1.

**PECAM1 N-terminus alone is sufficient for interaction.** The disruption caused by Y713F could be explained by a direct role of PECAM1 C-terminus or a distance effect transmitted to its extracellular N-terminus. To specifically investigate the N-terminus, transmembrane and cytoplasmic regions of PECAM1 were replaced with plasma membrane-targeting sequence[59]. SYFP2 was engineered into the intracellular side, generating PECAM1-ex-SYFP2, which localises to plasma membrane as expected (Fig. 4i). PIEZO1-mTurquoise2 and PECAM1-ex-SYFP2 enrich at cell-cell junctions (Fig. 4j) and donor (mTurquoise2) lifetime is affected exclusively at cell-cell junctions (Fig. 4k, l). Green fluorescent protein (GFP) trap was also used to

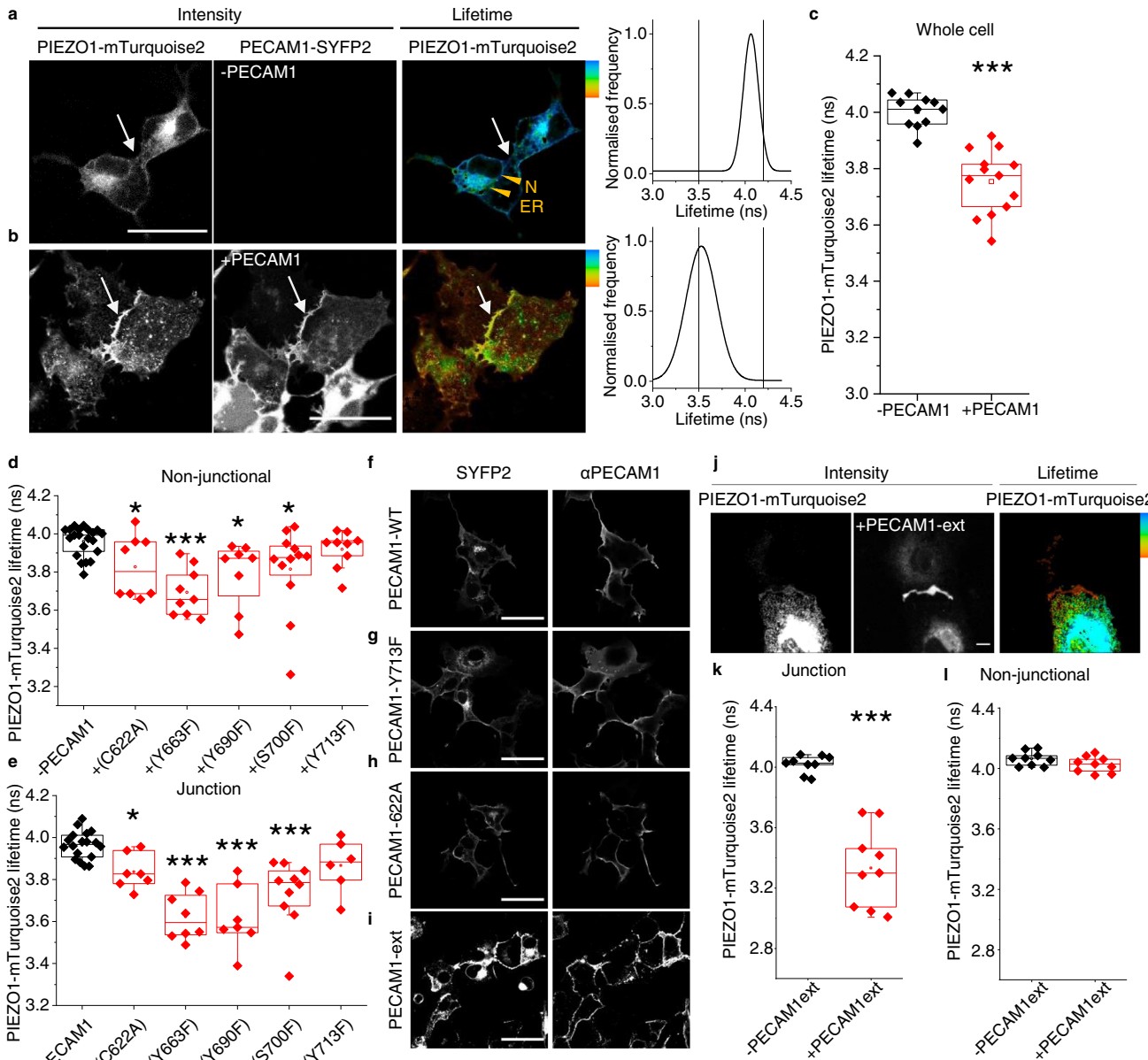

**Fig. 4 PECAM1 drives PIEZO1 to cell-cell junctions.** Data are for non-permeabilised COS-7 cells expressing human PIEZO1-mTurquoise2 with human PECAM1-SYFP2. **a–l** FRET/FLIM studies. **a** PIEZO1-mTurquoise2 alone. The intensity image in which white is high intensity and lifetime image calibrated to the rainbow scale (3.5–4.2 ns). The white arrows point to a region of cell-cell contact and the orange arrowheads nucleus (N) and endoplasmic reticulum (ER). The graph on the right is the lifetime distribution with the limits of the rainbow scale indicated by vertical lines. **b** Similar to **a** except PIEZO1-mTurquoise2 with PECAM1-SYFP2 (+PECAM1). **a**, **b** Scale bars, 50 μm. **c** Box plot presentation of summary data of the type shown in **a** and **b**, showing peak lifetime for the entire cell (Whole cell). $n = 4$ independent experiment repeats for PIEZO1-mTurquoise2 alone (−PECAM1, black) and PIEZO1-mTurquoise2 plus PECAM1-SYFP2 cells (+PECAM1, red). ***$P = 3.16 \times 10^{-5}$. The superimposed data points are average lifetimes for different images, −PECAM1 ($N = 11$) and +PECAM1 ($N = 13$). **d–h** Data are for COS-7 cells expressing PIEZO1-mTurquoise2 only (−PECAM1) or co-expressing PIEZO1-mTurquoise2 and PECAM1-SYFP2 in which PECAM1 was mutated at the indicated amino acid residue. Box plot presentations of FRET/FLIM peak lifetime data measured at: **d** Non-junctional (intracellular) regions (*$P_{C622A} = 0.0381$, ***$P_{Y663F} = 5.98 \times 10^{-4}$; *$P_{Y690F} = 0.0431$, *$P_{S700F} = 0.02136$, $P_{Y713F} = 0.13532$); **e** Cell-cell junction regions (*$P_{C622A} = 0.014$, ***$P_{Y663F} = 3.56 \times 10^{-4}$, ***$P_{Y690F} = 7.76 \times 10^{-4}$, ***$P_{S700F} = 2.07 \times 10^{-4}$, $P_{Y713F} = 0.1024$). Data are for $n = 3$ independent experiment repeats. The superimposed data points are average lifetimes for different images, −PECAM1 ($N_{non junctional} = 21$ and $N_{junctional} = 18$), +C622A ($N_{non junctional} = 8$ and $N_{junctional} = 7$), +Y663F ($N_{non junctional} = 9$ and $N_{junctional} = 8$), +Y690F ($N_{non junctional} = 8$ and $N_{junctional} = 7$), +S700F ($N_{non junctional} = 12$ and $N_{junctional} = 10$), +Y713F ($N_{non junctional} = 9$ and $N_{junctional} = 6$). Confocal images showing sub cellular distribution of **f** wild-type (WT) PECAM1, **g** Y713F and **h** C622A as SYFP2 fluorescence and after immunostaining using antibody to PECAM1 extracellular domain (αPECAM1) in non-permeabilised cells. Images are representative of $n = 3$ independent experiments. Scale bars, 50 μm. **i–l** Data are for COS-7 cells expressing PIN-G-tagged N-terminal PECAM1-ex-SYFP2 alone (PECAM1-ex) or with PIEZO1-mTurquoise2. **i** Confocal image showing sub cellular distribution of PECAM1-ex. **j** Intensity images and lifetime image calibrated to the rainbow scale indicated at the top right corner (3.5–4.2 ns). Scale bar, 50 μm, applies to all images. Box plot presentations of FRET/FLIM peak lifetime data measured at: **k** Cell-cell junctions (***$P = 4.12 \times 10^{-4}$); **l** Non-junctional regions ($P = 0.18533$). Data are for $n = 3$ independent experiment repeats. The superimposed data points are average lifetime values junctions (−PECAM1ext, $N = 9$ and +PECAM1ext, $N = 9$) and non-junctions (−PECAM1ext, $N = 9$ and +PECAM1ext, $N = 9$) from separate images.

bind PECAM1-SYFP2 (SYFP2 is a GFP variant) and test for coprecipitation of PIEZO1 co-expressed in HEK 293 cells. PIEZO1 is detected strongly when PECAM1-SYFP2 is coexpressed, suggesting precipitation that depends on PECAM1 (Supplementary Fig. 13). Consistent with a dominant role of N-terminal PECAM1, the Y713F mutation does not affect the precipitation (Supplementary Fig. 13). The data suggest that N-terminus of PECAM1 is sufficient for PIEZO1 localisation to junctions and its PECAM1 interaction.

**Preference for hypoglycosylated PECAM1**. PECAM1 migrates at multiple molecular masses and the isoform with the smallest mass coprecipitates with PIEZO1 (Supplementary Fig. 14a). Post-translational hypoglycosylation of PECAM1 occurs at mature cell-cell junctions to improve the strength of transhomotypic interactions[60,61]. Treatment with N-Glycosidase F (PNGase F) confirms that the smallest molecular mass band is the hypoglycosylated state (Supplementary Fig. 14b). The data suggest that PIEZO1 preferentially interacts with hypoglycosylated PECAM1, the species of mature junctions.

**CDH5 also drives PIEZO1 to cell-cell junctions**. To determine if the other components of the PECAM1 triad, CDH5 and VGFR2, couple with PIEZO1, we studied CDH5-mVenus and VGFR2-SYFP2 in COS-7 cells. mVenus and SYFP2 differ by one amino acid and have similar spectra. Like PECAM1, CDH5 drives PIEZO1 to cell-cell junctions (Fig. 5a) and reduces the lifetime of the donor fluorophore (mTurquoise2), suggesting interaction (Fig. 5a, c, and Supplementary Fig. 15a–e). VGFR2-SYFP2, however, lacks effect on PIEZO1 localisation (Fig. 5b *cf* Fig. 4a) or mTurquoise2 lifetime (Fig. 5d). Similar to the finding with PECAM1 (Supplementary Fig. 14), PIEZO1 preferentially coprecipitates with hypoglycosylated CDH5 (Supplementary Fig. 16). To investigate the relevance to the native proteins, we returned to retinal vascular studies. Staining of retinas of PIEZO1[HA] mice shows colocalisation of endogenous CDH5 and PIEZO1 in situ in retinal vein (Fig. 5e–j). The data suggest that CDH5, but not VGFR2, also drives PIEZO1 to cell-cell junctions and interacts with it.

**Shear stress enhances CDH5 but not PECAM1 interaction**. CDH5 is not equivalent to PECAM1 in the triad. It is recruited as an adaptor in response to shear stress[10]. We therefore investigated the effect of shear stress on PIEZO1-related FRET/FLIM signals in COS-7 cells. Preconditioning shear stress induced by fluid flow established a physiological cell condition prior to a static no-flow period. Then shear stress was applied again for 10 min or cells were retained in static condition (Fig. 6a–j). Shear stress has no effect on the ability of PECAM1-SYFP2 to lower the lifetime of the donor fluorophore of PIEZO1 (Fig. 6a–e) but increases the effect of CDH5-mVenus specifically at cell-cell junctions (Fig. 6f–j). VGFR2-SYFP2 showed no FRET signal, with or without shear stress (Supplementary Fig. 17). The data suggest that shear stress increases PIEZO1's interaction with CDH5 but not PECAM1, consistent with additional CDH5 being recruited in response to force.

**CDH5 lacks effect on PIEZO1 channel activity**. CDH5 interacts with PIEZO1 and so we studied its effects on PIEZO1 channel activity using HEK 293 cell patch-clamp recording. However, no effects on PIEZO1 channel currents are evident (Supplementary Fig. 18). The data suggest that CDH5 lacks effect on PIEZO1 activity, despite its interaction.

**Physiological PIEZO1 increases junction width and radial actin in endothelial cells**. CDH5 is $Ca^{2+}$ regulated and so PIEZO1 channels could serve to regulate $Ca^{2+}$ locally and thereby link local mechanical force to adherens junction structure. Consistent with this hypothesis, force-induced junctional remodelling in HUVECs is associated with transient elevation of cytosolic $Ca^{2+}$ concentration sufficient for junctional remodelling[46]. We depleted and then reintroduced extracellular $Ca^{2+}$ to observe $Ca^{2+}$ regulated junction formation in confluent HUVECs (Fig. 7a), testing the role of PIEZO1 by depleting it (but not PECAM1 or CDH5) using PIEZO1 targeted siRNA (Supplementary Fig. 19). In the PIEZO1-depleted group, there are thinner (tighter) junctions in the +calcium (post) condition (Fig. 7a–e, Supplementary Fig. 20). We also stained F-actin using phalloidin 568 (Supplementary Fig. 21a–c) because cytoskeletal architecture is coordinated with junctional remodelling and affected by stretch[46]. In the PIEZO1-depleted condition, there are fewer F-actin peaks in cross-section, suggesting less radial actin (which spans focal adhesions) and more cortical actin (which coordinates with cell junctions). This is most apparent when extracellular $Ca^{2+}$ is returned after $Ca^{2+}$ depletion (+calcium (post) in Supplementary Fig. 21d–f). The data suggest that PIEZO1 increases the width of junctions, consistent with the junctions being less tight and more able to remodel. Moreover, PIEZO1 promotes radial actin, which is also consistent with PIEZO1 facilitating cell and junctional remodelling (Fig. 7f, g).

**Pharmacological activation of PIEZO1 causes radial actin collapse**. External force was not applied in the studies of Fig. 7 and Supplementary Fig. 21 and so we assume PIEZO1 was activated only by physiological forces inherent to the cells and their substrate. Application of the small-molecule agonist of PIEZO1, Yoda1, near its concentration for 50% effect $(3\,\mu M)$[62] causes substantial intracellular $Ca^{2+}$ elevation above basal levels of such cells, suggesting strong additional PIEZO1 activation (Supplementary Fig. 19a, b). This concentration of Yoda1 strikingly disorganises F-actin structure (Supplementary Fig. 22). Something similar may have occurred when Yoda1 was infused in situ (Fig. 2d, e).

## Discussion

From these results, we suggest connection of PIEZO1 and PECAM1 concepts in endothelial force sensing through protein-protein interaction and a pool of PIEZO1 at cell junctions in addition to the already established pool at the apical membrane[1,38,39,47] (Fig. 8). We show similarity of PIEZO1's in vivo expression pattern to that of PECAM1, an established junctional protein. We show PIEZO1's functional suppression when the amount of PECAM1 increases and that PECAM1 inhibits PIEZO1's mechanical sensitivity, potentially through constraint of its N-terminus, the force sensing region. We show PIEZO1 and PECAM1 reconstitution at cell-cell junctions and that PECAM1 drives PIEZO1 to junctions. PECAM1 N-terminus is sufficient for interaction but intracellular C-terminal regions previously linked to sensing of shear stress also participate. CDH5, the other cell adhesion molecule of PECAM1's triad, similarly drives PIEZO1 to junctions and interacts with it, in this case regulated by shear stress.

Determining specific roles of the junctional pool will be challenging but PIEZO1 channels are highly adapted to sensing increased membrane tension[52], so we suggest a role of this PIEZO1 in detecting local tension in junctional membranes, coupling it to junctional structures via local $Ca^{2+}$ signalling. We show that PIEZO1 is required for $Ca^{2+}$-dependent remodelling of junctions and associated actin cytoskeleton. Interaction of

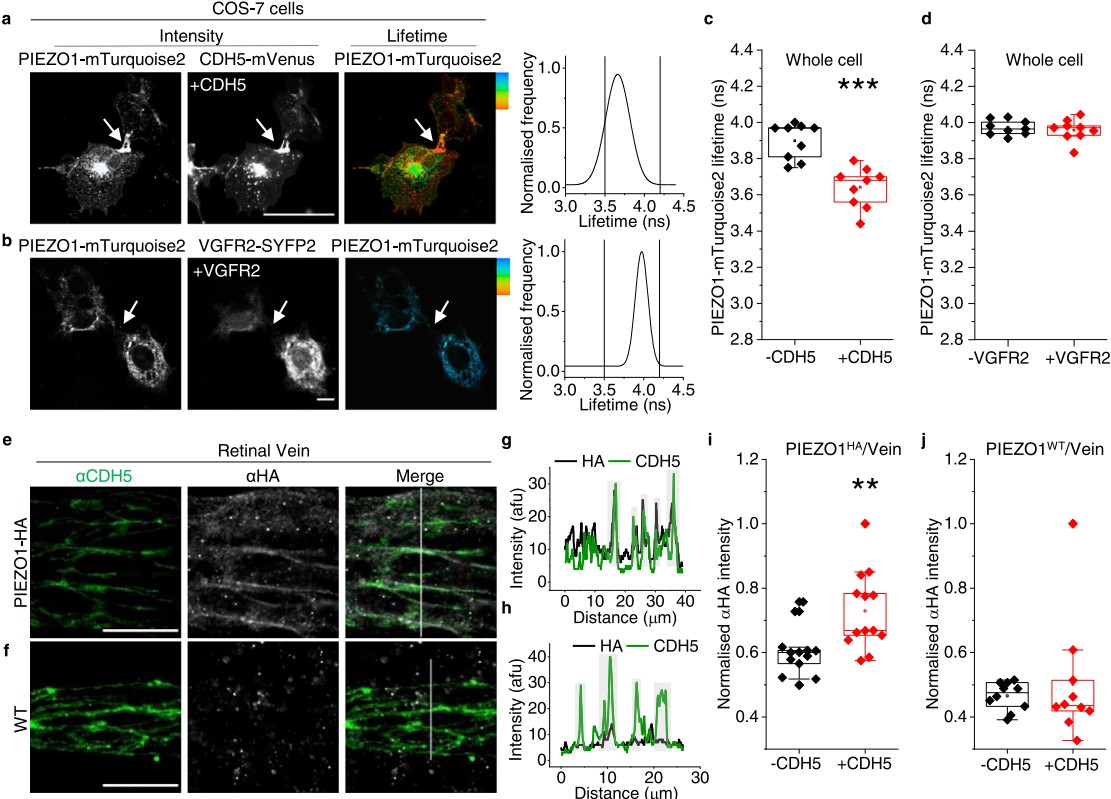

**Fig. 5 CDH5 but not VGFR2 partners with PIEZO1. a–d** FRET/FLIM images and analysis for COS-7 cells expressing **a** PIEZO1-mTurquoise2 plus CDH5-mVenus (CDH5-mVenus) or **b** PIEZO1-mTurquoise2 plus VGFR2-SYFP2. **a, b** Intensity images (white, high intensity), lifetime images calibrated to the rainbow scale indicated at the top corner (3.5–4.2 ns) and graphs of the lifetime distributions in which grey vertical lines indicate the rainbow scale limits. Scale bars, 50 μm, apply to all images. **c, d** Box plot summary peak whole cell lifetime data for the experiment types of **a** and **b**. 3 independent experimental repeats for PIEZO1-mTurquoise2 alone (-CDH5/VGFR2, black) and PIEZO1-mTurquoise2 plus CDH5-mVenus (***$P = 7.68 \times 10^{-4}$) or VGFR2-SYFP2 ($P = 0.72393$) (+CDH5/VGFR2, red). Superimposed data points are average lifetime values for individual images ($-$CDH5, $N = 9$; $+$CDH5, $N = 9$; $-$VEFGR2, $N = 9$; $+$VGFR2, $N = 9$). **e–j** Images and image analysis for retinal veins of HA-PIEZO1 mice (PIEZO1^HA) or wild-type (WT) mice (PIEZO1^WT) immuno-stained with αCDH5 antibody (green) and αHA antibody (grey). **e, f** Representative images for αCDH5 and αHA staining with merger of these images to the right. Scale bars, 20 μm. **g, h** are the line-intensity (grey value) plots for the vertical scan lines superimposed on the Merge images of **e** and **f**. Green αCDH5, grey αHA. Light grey highlighting indicates cell-cell junctions. **i, j** Box-plot quantification of image intensity for **e** PIEZO1^HA or **f** PIEZO1^WT retinal veins stained with αHA, shown for junctional regions indicated by αCDH5 staining (+CDH5) and non-junctional regions (−CDH5) in arbitrary fluorescence units (AFU). The intensity of each image was normalised to the image background. **$P = 0.00347$. Data are for $n = 3$ independent experiments. The superimposed data points are the average intensity for individual images, PIEZO1^HA (−CDH5, $N = 13$ and +CDH5, $N = 13$) PIEZO1^WT (−CDH5, $N = 10$ and +CDH5, $N = 10$).

PIEZO1 with PECAM1 and CDH5 may be important in enabling such remodelling to happen efficiently. Whether both PIEZO1 and the adhesion molecules sense force is unknown, but we note the compelling evidence that PIEZO1 channels are direct and specific sensors of mechanical force, apparently having evolved for this purpose[25–27], including in endothelial cells[30,38,39,63]. Therefore, without excluding mechanical detection by the PECAM1 triad, we suggest that critical force sensing arises at PIEZO1 channels and that they may confer force sensitivity on the triad.

Interaction between PIEZO1 and cell adhesion molecules could exist simply to enable PIEZO1 to reach junctions but functional interaction occurs too. We suggest negative feedback from PECAM1 to PIEZO1 when junctional density becomes high, serving to dampen PIEZO1's remodelling role. In addition, we envisage Ca$^{2+}$ permeability of junctional PIEZO1 channels regulating the local cytosolic Ca$^{2+}$ concentration under the junctional membrane, thereby locally activating Ca$^{2+}$-dependent mechanisms such as calpain to regulate junctional organisation[29,30,42]. Calpain is a known downstream mediator of PIEZO1 effects, regulator of cytoskeletal anchorage complexes

and component of the endothelial shear stress sensing machinery[29,30,42,43,64–66]. PECAM1 is cleaved by calpain[67]. Apical PIEZO1 may be largely independent of junctional proteins such as PECAM1 because PECAM1 is primarily at junctions in endothelial cells. It is nevertheless important, detecting shear stress at the apical surface[38,39] and triggering distance signalling. The detection may involve intermediates[40,41] and signalling via Ca$^{2+}$ elevation, ATP release, G protein-coupled receptors (e.g., P2Y$_2$ receptors), phospholipase Cβ and other systems to coordinate apical shear stress with junctional structure and other events such as the production of nitric oxide[1,42,47].

Ca$^{2+}$-permeable channels regulate cytosolic Ca$^{2+}$ but depletion of local extracellular Ca$^{2+}$ could also be relevant here because of the restricted extracellular space of endothelial cell-cell junctions. A case has been made for local depletion of extracellular Ca$^{2+}$ at similarly-sized synaptic junctions[68]. Depletion of extracellular Ca$^{2+}$ in narrow diffusion-restricted spaces between endothelial cells, which have a width about the size of a PIEZO1 channel[33] (15–30 nm)[69], may be sufficient to cause Ca$^{2+}$ to dissociate from ectodomains of PECAM1 and CDH5, a consequence of which is expected to be less transhomophilic interaction and

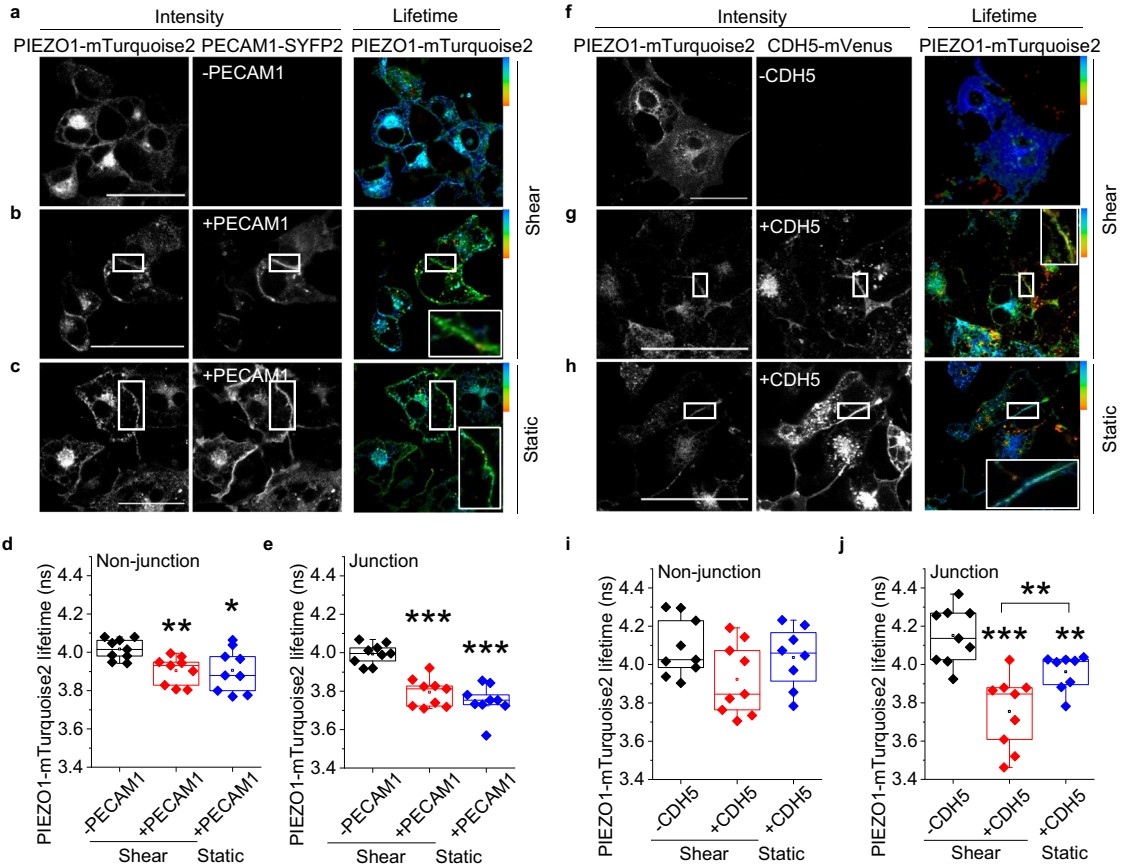

**Fig. 6 CDH5 partnering is shear stress dependent.** COS-7 cell data obtained by FRET/FLIM after expressing PIEZO1-mTurquoise2 with (+) or without (−): **a–e** PECAM1-SYFP2 (PECAM1); **f–j** CDH5-mVenus (CDH5). Prior to imaging, cells were preconditioned for 24 h with laminar shear stress (10 dyn.cm$^{-2}$) followed by static condition for 30 min and then 10 min 10 dyn.cm$^{-2}$ (shear) or continued static condition (static). For 3 independent experiments each, the box plots show the lifetimes for PIEZO1-mTurquoise2 at cell-cell junctions: **e** +PECAM1 ***$P = 2.64 \times 10^{-4}$ shear and ***$P = 1.37 \times 10^{-4}$ static, both compared with −PECAM1; **j** +CDH5 ***$P = 2.63 \times 10^{-4}$ shear and **$P = 0.0079$ static, both compared with −CDH5. +CDH5 shear cf +CDH5 static **$P = 0.00036$ and the lifetimes for PIEZO1-mTurquoise2 at non junctional regions: **d** +PECAM1 **$P = 0.0021$ shear and *$P = 0.011$ static, both compared with −PECAM1; **i** +CDH5 showed no significant differences to the −CDH5 under shear and static conditions. The superimposed data points are for individual junctions ($N = 9$) and non-junctions ($N = 9$) from separate images. P values are from Mann–Whitney tests with Bonferroni correction.

weaker cell-cell contact, facilitating junction remodelling. CDH5 is an inherently Ca$^{2+}$ sensitive protein, belonging to an extended family of Ca$^{2+}$ binding adhesion molecules[21]. The Ca$^{2+}$ binding occurs in triplicates at multiple sites in the extracellular N-terminus and is important for the rigid crescent shape of the extracellular domain, the structure of X-dimer intermediate and opening of the A strand[21]. PECAM1 also contains extracellular Ca$^{2+}$ binding sites that are more restricted, but at a position associated with modulated homophilic binding affinity[22,70]. Such Ca$^{2+}$ binding sites are thought to be saturated at plasma Ca$^{2+}$ concentrations but the Ca$^{2+}$ affinity is relatively low and so the possibility exists for Ca$^{2+}$ unbinding, should extracellular Ca$^{2+}$ decline[68]. This supposition is encouraged by our finding that the extracellular domain of PECAM1 interacts with PIEZO1 at cell-cell junctions. Therefore, binding of the PECAM1 extracellular domain in the vicinity of the outer vestibule of the PIEZO1 channel could orchestrate a local sink-like effect in which Ca$^{2+}$ is efficiently drawn away from the PECAM1 extracellular domain. Consistent with this idea, our data point to preferential interaction of PIEZO1 with hypoglycosylated states of PECAM1 and CDH5, suggesting optimisation for mature junctions, which are achieved partly by posttranslational deglycosylation[60,61].

We identified importance of tyrosine 713 (Y713) in PECAM1's relationship with PIEZO1. Deletion of a domain containing this residue prevents tyrosine phosphorylation of PECAM1 in response to mechanical force[9]. This domain may also associate reversibly with the inner leaflet of the bilayer, controlling its phosphorylation[71]. The adjacent exon 13- and 15- encoded domains are the suggested points of interaction of PECAM1 with γ-catenin and β-catenin, conferring physical links to the cytoskeleton[72]. These findings point to additional mechanisms by which a PIEZO1-PECAM1 partnership could confer integration and regulate force sensing. Moreover, our data highlight the value of protein-protein interaction studies in situ in cells using techniques such as FRET/FLIM where the 3-dimensional subcellular architecture and localisation mechanisms of cells are retained. Our biochemical approaches that dissipated cell structure did not reveal an effect of mutating Y713.

CDH5's association with PIEZO1 is enhanced by shear stress, suggesting that it has a more dynamic relationship with PIEZO1 than PECAM1. Evidence for diversity of PIEZO1 relationships with cell adhesion molecules is emerging. In epithelial cells, CDH1 interacts with PIEZO1. CDH1 has a strong enhancing effect on PIEZO1 channel function[73]. This study also reported an enhancing effect of CDH5 on PIEZO1 channel activity, albeit

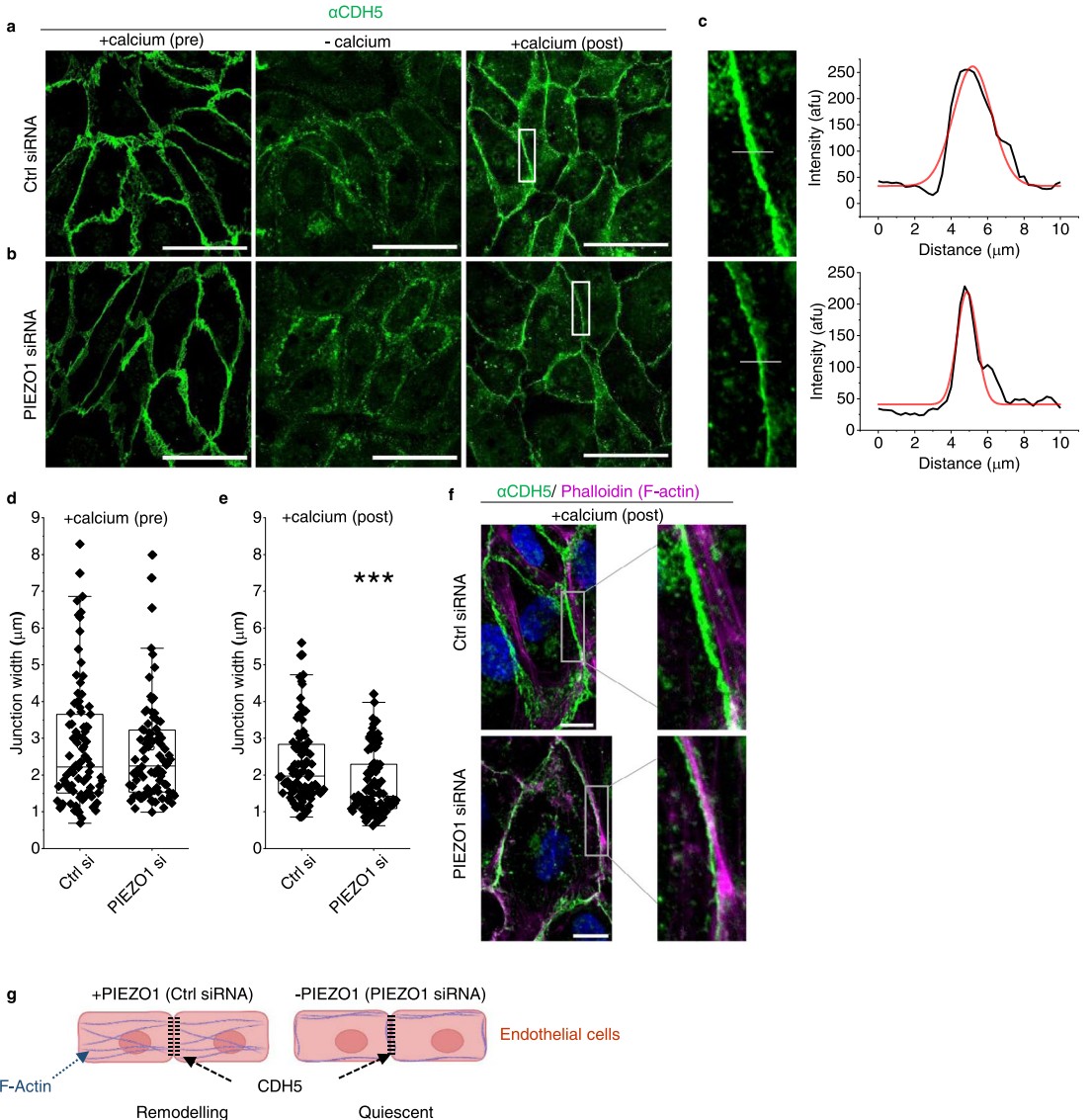

**Fig. 7 PIEZO1 increases junction width and radial actin.** HUVEC data. **a–f** Cells cultured in a confluent monolayer treated with **a** control (Ctrl) or **b** PIEZO1 siRNA and subjected to extracellular $Ca^{2+}$ switch assay: pre-treatment (normal $Ca^{2+}$); −calcium (30 min $Ca^{2+}$-depletion); and recovery (+calcium (post); 30 min after restoring $Ca^{2+}$). After treatments, cells were stained with anti-CDH5 antibody (αCDH5) (green). The scale bars are 100 μm. **c** Images are enlarged from the white boxes in **a**, **b** +calcium (post). To the right are line-intensity plots (black) with a Gaussian fit (red) used to calculate peak width at half maximum for the superimposed grey lines shown in the enlarged images. **d**, **e** As for **a** and **b** but showing box plots for +calcium pre-treatment ($n = 3$, $P = 0.92988$) and +calcium (post) ($n = 3$, ***$P = 8.86242 \times 10^{-5}$) conditions. Superimposed data points are measurements from individual cell-cell junctions for +calcium (pre) (Ctrl si = 89, PIEZO1 si = 90) and +calcium (post) (Ctrl si = 87, PIEZO1 si = 88). **f** Enlarged and merged images of CDH5 (green, from **a**, **b**) and F-actin (phalloidin) (magenta, from Supplementary Fig. 21a, b) staining for the calcium switch recovery (+calcium (post); 30 min after restoring $Ca^{2+}$) conditions in Ctrl or PIEZO1 siRNA treated HUVECs. Scale bars are 25 μm **g** Schematic representation of F-actin and CDH5 with (+) and without (−) PIEZO1 or PIEZO1 activation by mechanical force, based on the data of **a–f** and Supplementary Fig. 21. In the +PIEZO1 condition, there is suggested to be junctional remodelling with more radial actin and wider (less tight and more leaky) junctions.

smaller than that of CDH1. The technique used for mechanical activation was 'cell poking' with a stylus, which may not be comparable with the membrane stretch applied in our experiments. Nevertheless, the data are consistent with our proposal that CDH5 does not inhibit PIEZO1 function.

PIEZO1 channels are also linked to NOTCH1[63], which participates in cell-cell interaction through NOTCH ligands in endothelial cells and other cell types[74]. ADAM10 sheddase, which mediates the effect of PIEZO1 on NOTCH1[63], cleaves CDH5[75]. Membrane-bound NOTCH1 signals to CDH5 via Rac1 to drive assembly of adherens junctions[76]. These observations suggest a broader role of PIEZO1 in endothelial cell-cell junctions beyond PECAM1 and CDH5 and add to an emerging picture of PIEZO1 channels as mechanical detectors with important roles in cell-cell interactions.

The endothelial response to shear stress may often occur alongside adaptive changes in endothelium such as cytoskeletal rearrangements and junction remodelling in diapedesis, endothelial remodelling, inflammation or other events. PIEZO1 is important here too, apparently existing as an adaptable mechanical force-sensing cassette in multiple subcellular compartments. PIEZO1 stimulation straightens CDH5 junctions and

**Fig. 8 Model for PIEZO1 and PECAM1 partnership.** Sketch of part of two adjacent endothelial cells with a cell-cell junction in between. The junction is 15–30 nm in reality and so much narrower than apparent in the sketch. The junction includes adherens and tight junctions, but only the adherens junction is referred to here. Two pools of PIEZO1 channels are proposed. One pool is in the apical membrane of the endothelial cell and suggested to be particularly involved in sensing force as part of the shear stress sensing machinery. The other pool is in the adherens junction membrane of the endothelial cell and suggested to be particularly involved in sensing forces such as membrane tension as part of the adherens junction force sensing machinery. Force sensing is suggested to be mediated by PIEZO1 channels in both cases, leading to local and distance signalling to modulate endothelial cell function. Integration between the pools is envisaged to coordinate apical and junctional membrane events. PIEZO1 channels are $Ca^{2+}$-permeable non-selective cation channels and so local intracellular $Ca^{2+}$ elevation and extracellular $Ca^{2+}$ depletion are likely when the channels open and this may contribute to regulation of nearby mechanisms, such as F-actin and the adhesion molecules PECAM1 and CDH5. We suggest also direct interaction between PIEZO1 and the adhesion molecules that is important both for localising PIEZO1 to junctions and regulating junctional structure once PIEZO1 is at the junctions. Negative feedback is suggested to occur from PECAM1 to PIEZO1 as junctional intensity increases to enable PIEZO1's role in driving junctional remodelling to be suppressed once remodelling is complete and junctions need to return to a tighter, less leaky, state.

its depletion inhibits stretch-evoked remodelling[45,46]. PIEZO1 also has roles in focal adhesions[30,77], to which radial actin is attached, consistent with a remodelling role of PIEZO1. Although our data and that of other investigators support roles for PIEZO1 as an enabler of junctional remodelling, whether this results in tighter or weaker junctions (less or more permeability) may depend on context as PIEZO1 depletion enhances or suppresses vascular permeability in vivo[42,43].

We bring apparently competing ideas closer together but there remain major questions to answer about how endothelium responds to shear stress and other mechanical forces. We would like to know, for example, the first event or events when shear stress increases and the downstream consequences as the coordinated system response unfolds. We would like to know how these processes vary depending on the type of blood vessel or lymphatic, the context, the direction and chaos of the shear stress and the co-applied forces such as membrane tension and radial stretch. We are a long way from fully understanding. Instead, we know some molecules that are critical and a plethora of molecular changes that occur over time. What we lack is a time-series of events and their spatial orchestration. A key idea emerging from our work and that of others is that PIEZO1 channels are extremely rapid and sensitive responders to mechanical force sitting at the heart of this biology. Our hypothesis is that activation of these channels is a first or near-first event when force is applied at the endothelial surface or glycocalyx. This puts PECAM1

downstream. Consistent with this model, tyrosine phosphorylation of PECAM1 measured 5 min after the start of shear stress depends on PIEZO1[32]. However, we need much more sophisticated experiments to understand what happens. Our findings suggest an approach in which it would be necessary to measure in real-time the protein structural conformations of PIEZO1 and PECAM1 at apical and junctional membranes during shear stress. This would ideally be complemented by experiments incorporating specific disruption of the PIEZO1-PECAM1 interaction, thereby enabling testing of its role and the role of the proposed negative feedback of PECAM1 on PIEZO1.

Some of our conclusions are based on expression of tagged proteins in model cell systems, including non-endothelial cells so that we could utilise their technical advantages and reconstitute effects on null backgrounds for PIEZO1, PECAM1 and CDH5. Such technical approaches confer accuracy and reliability, are flexible, can be robustly tested and controlled and enable the relatively simple testing of the roles of specific amino acid residues through expression of mutants. However, there are potential limitations. The host cell type may not handle the exogenous proteins in the same way as the native cell. The experiments may involve overexpression (i.e., abnormally high protein expression that does not occur physiologically and which may cause effects that are not physiological). We took care to minimise the risk of abnormal expression, using transfection efficiencies that were just sufficient to enable detection of tagged constructs in natural

cellular patterns. We also used multiple methods and cell types to validate our conclusions as each system has different limitations. The arising data provide a framework for designing future studies of such mechanisms in vivo, for example through the engineering of mice containing mutant or fluorescently-tagged endogenous proteins that can be studied in the native context of endothelial cells.

The suggested inhibitory effect of PECAM1 on PIEZO1 function is based both on our overexpression studies in HEK 293 cells and studies of the native proteins in endothelial cells under conditions of high PECAM1 abundance and junctional intensity. These studies independently point to the same conclusion and are consistent with a prior suggestion that degradation of PIEZO1 facilitates junctional normalisation after stretch-evoked remodelling[46]; i.e., that PIEZO1 activity is enabled for remodelling but otherwise suppressed. The mechanism of this inhibitory effect of PECAM1 on PIEZO1 channel activity is unknown. It could relate to an effect of PECAM1 on membrane stiffness, local protein density or local lipid composition, although the effect was not reproduced by CDH5, which might be expected to similarly produce such disturbances. We successfully copurified PIEZO1 and PECAM1 overexpressed in HEK 293 cells, consistent with there being interaction, but this should be seen in the context of the central idea that there are two pools of PIEZO1 – one associated with and one not associated with PECAM1. We suggest that the associated pool is at cell-cell junctions, which are small but important. We investigated this pool in situ using FRET/FLIM and based on the arising data suggest stable association at cell-cell junctions of quiescent cell monolayers after junctions have formed. While the results of the HEK 293 cell experiments support our hypothesis, the value of this approach is limited in the context of such complex biology. Investigation at the site where the association occurs is especially important. Future structural studies may reveal details of the interaction that can then be used to understand it and enable its specific disruption in the native endothelial context.

We show that pharmacological activation of PIEZO1 (by the agonist Yoda1) disrupts PECAM1's structural organisation and causes the collapse of radial actin. Our data suggest that Yoda1 can activate PIEZO1 substantially above its physiological activation levels, generating effects that are not necessarily physiologically relevant. In future studies, titrating Yoda1's concentration down may lead to better mimicry of PIEZO1 activation by physiological force, although it may also be necessary to apply the Yoda1 in a localised and directional manner, as occurs with forces such as shear stress and membrane tension.

Our findings help to resolve an apparent contradiction in understanding how force is sensed by endothelium, placing PIEZO1 at the centre of an integrated concept across subcellular compartments of the endothelium. In the future, it will be interesting to investigate the mechanisms in more detail and elucidate their specific in vivo relevance. It will also be interesting to determine if relationships of PIEZO1 to PECAM1 and CDH5 extend to other cell adhesion molecules and other cell types. PIEZO1 is widely expressed[26,29] and there are numerous cell adhesion molecules in endothelial cells and other cell types[21,23]. Studies of CDH1 already support the idea of broader relevance and suggest general roles of adhesion molecules in tuning PIEZO1's mechanical sensitivity and its relationship to cytoskeleton[73].

## Methods

**Cell lines**. COS-7 cells were from the American Type Culture Collection (ATCC) and maintained in Dulbecco's Modified Eagle Media (DMEM) supplemented with 10% FCS, 2 mM L glutamine, 100 U·ml$^{-1}$ penicillin, 100 μg·ml$^{-1}$ streptomycin in 5% CO$_2$, 95% air atmosphere. Cells were sub-cultured upon reaching a surface area density of 80–90% by detaching with 0.5% trypsin. T-REx-293 cells from Thermo Fisher Scientific engineered with tetracycline inducible expression of human PIEZO1[38] were maintained in DMEM supplemented with 10% FCS, 2 mM L-glutamine, 100 U·ml$^{-1}$ penicillin, 100 μg·ml$^{-1}$ streptomycin in 5% CO$_2$, 95% air atmosphere. Cells were selected with zeocin (400 μg·ml$^{-1}$) and blasticidin (5 μg·ml$^{-1}$) and induced with tetracycline (100 ng·ml$^{-1}$) for 24 h. T-REx-293 cells transfected with PECAM1 were generated by seeding at 70% confluency in a 6-well plate. Cells were transfected for 5 h with 500 ng PECAM1 plasmid and 0.3% Lipofectamine 2000 (Invitrogen). 48 h after selection with 5 μg·ml$^{-1}$ blasticidin began, single cell lines stably expressing PECAM1 were isolated. HUVECs were cultured in EGM-2 growth medium supplemented with EGM-2 bullet kit (Lonza). Cells were maintained at 37 °C in a humidified atmosphere containing 5% CO$_2$. T-REx-293 cells overexpressing mouse PIEZO1 were generated by seeding at 70% confluency in a 6-well plate. Cells were transfected for 5 h with 500 ng mPIEZO1 plasmid and 0.3% Lipofectamine 2000 (Invitrogen). 48 h after selection with 5 μg·ml$^{-1}$ blasticidin began, single cell lines stably expressing mPIEZO1 were isolated.

**Mouse breeding and husbandry**. All animal use was authorised by the University of Leeds Animal Ethics Committee and The Home Office, UK. Animals were maintained in GM500 individually ventilated cages (Animal Care Systems) at 21 °C, 50–70% humidity, light/dark cycle 12/12 h on chow diet ad libitum and bedding of Pure'o Cell (Special Diet Services, Datesand Ltd, Manchester, UK). Genotypes were determined using real-time PCR with specific probes designed for each gene (Transnetyx, Cordova, TN).

**Generation of HA-PIEZO1 mice C57BL/6J**. Mice with PIEZO1-HA tag were generated by introducing an HA sequence between amino acids A2439 and D2440 by CRISPR-Cas9 (Fig. 1 Supplementary Fig. S1). A sgRNA was selected based on proximity to the target region and low off targeting potential (*catgcgagctgcaggactgca-agg*; (https://www.sanger.ac.uk/htgt/wge/crispr/538567112), and a ssDNA repair template with the HA tag sequence and 60nt flanking homology arms was designed to facilitate integration of the HA tag sequence after Cas9 induced double strand break was synthesised (Integrated DNA technologies, with PAGE purification). sgRNA sequence was synthesised as an Alt-R crRNA (Integrated DNA Technologies) oligo and re-suspended in sterile Opti-MEM (Gibco) and annealed with tracrRNA (Integrated DNA Technologies) by combining crRNA (2.5 μg) with tracrRNA (5 μg) and heated to 95 °C. After annealing the complex, an equimolar amount was mixed with Cas9 recombinant protein (1500 ng) (NEB), the ssDNA repair template (final concentration 10 ng·μl$^{-1}$) in Opti-MEM (total volume, 15 μl) and incubated (RT, 15 min). Mouse embryos were electroporated (Nepa21 electroporator, Sonidel) using AltR crRNA:tracrRNA:Cas9 complex (200 ng·μl$^{-1}$; 200 ng·μl$^{-1}$; 200 ng·μl$^{-1}$ respectively) and ssDNA HDR template (500 ng·μl$^{-1}$)[78]. Zygotes were cultured overnight and the resulting 2 cell embryos surgically implanted into the oviduct of day 0.5 post-coitum pseudopregnant mice. After birth and weaning genomic DNA extracted using REDExtract-N-Amp™ tissue PCR kit (Sigma) and used to genotype pups by PCR using primers *cgactctaactatcc-cactcaac* and *atccctctgcagtactcacc*, followed by Sanger sequencing of candidate pup1. Mice were bred to obtain homozygotes for HA-tag-PIEZO1 (PIEZO1$^{HA}$).

**DNA constructs and cloning**. human PIEZO1-mTurquoise2 was sub-cloned from human PIEZO1-GFP[30]. pSYFP2-C1[79] (Addgene plasmid # 22878; http://n2t.net/addgene:22878; RRID:Addgene_22878) was a gift from Dorus Gadella. mTurquoise2-C1[80] (Addgene plasmid # 54842; http://n2t.net/addgene:54842; RRID:Addgene_54842) was a gift from Michael Davidson and Dorus Gadella. Human PECAM1 was obtained from Origene (TrueClone, SC119894). PIEZO1 and PECAM1 pcDNA4 templates were generated by inverse PCR with the Phusion® DNA polymerase (New England Bio Labs). mTurquoise2 and SYFP2 were incorporated to the PIEZO1/PECAM1 templates by overlap PCR using In-Fusion® HD cloning kit (Takara). Linkers were then attached between PIEZO1/PECAM1 and mTurquoise2/SFYP2 using In-Fusion® HD cloning kit (Takara). For PIEZO1/mTurquoise2 and PIEZO1/SYFP2 the linker was flanked by a BamH1 restriction. The linker on PECAM1/SYFP2 and PECAM1/mTurquoise2 was flanked by a HindIII restriction site. VGFR2-SYFP2 was sub-cloned from mEmerald-VGFR2-N1 (Addgene plasmid # 54298; http://n2t.net/addgene:54298; RRID:Addgene_54298) was a gift from Michael Davidson. When sequenced, mEmerald-VGFR2-N1 was found to contain a frameshifting CT insertion between residues T1303-A1304 and the single nucleotide polymorphisms (SNPs) V297I and H472Q. Site-directed mutagenesis was performed to remove the insertion and correct the SNPs. To facilitate cloning of VGFR2-SYFP2, a linker flanked by AgeI and SacII restriction sites was introduced into pcDNA™4/TO between EcoRI and XhoI restriction sites using Gibson Assembly® (New England Biolabs). The SYFP2 fluorophore was inserted downstream of the linker between SacII and XbaI restriction sites using pSYFP2-C1. PCR products were assembled using Gibson Assembly® to insert VGFR2 upstream of the linker. The C-terminal fluorophore was removed from this construct using Gibson Assembly® to assemble VGFR2 and vector PCR products. mVenus-CDH5-N-10 (Addgene plasmid # 56340; http://n2t.net/addgene:56340; RRID:Addgene_56340) was a gift from Michael Davidson and was used as acquired. To generate C-terminal HA-tagged PIEZO1 the C-terminal

GFP from the human PIEZO1-GFP construct[30] was first removed by ligating the KpnI and NotI restriction enzyme digested vector and PIEZO1 PCR product. This construct was subsequently used as a template to create full-length and truncated PIEZO1 expression vectors. PCR products covering (T1) full length PIEZO1 sequence to L2471, G2174, I2089, S1591 and A1128 were produced and inserted into the vector using Gibson Assembly®. PIEZO1[HA] (CED). An HA-tag was also introduced between T2413 and C2414 located in the C-terminal extracellular domain. PECAM1 mutants were generated using the PECAM1-SYFP2 plasmid as a template. Mutagenesis was carried out using PrimeSTAR HS DNA polymerase (TaKaRa). The extracellular domain of PECAM1 containing SYFP2 and PIN-G was cloned from the PECAM1-SYFP2 plasmid using In-Fusion® HD cloning kit (Clonetech). The human PIEZO1-GFP construct used for co-purification was generated from PIEZO1-GFP[30] using round-the-horn PCR to insert a HRV3C protease cleavage site between PIEZO1 and the GFP tag. Halo tagged PIEZO1 was obtained from the Kazusa DNA Research Institute. pcDNA3_mouse PIEZO1_IRES_GFP, (Addgene plasmid # 80925; https://www.addgene.org/80925) a gift from A Patapoutian, was used as a template to clone the mouse PIEZO1 (mPIEZO1) coding sequence, into pcDNA™4/TO. Overlapping mouse PIEZO1 and pcDNA™4/TO PCR products (PrimeSTAR HS DNA Polymerase, TaKaRa) were assembled using Gibson Assembly (NEB). Constructs did not contain tetracycline operator sequences.

**Paraformaldehyde fixation.** Cells were fixed with 4% paraformaldehyde (10 min, RT) and washed with PBS 3 times for 5 min. Fixation was quenched with glycine (0.1 M in PBS, 10 min, RT) and cells were washed with PBS (3 times, 5 min, RT). Fixed cells were covered in PBS and stored at 4 °C. All fixation steps were carried out under low light and sterile condition. Samples were imaged on the same day or stored for a maximum of 24 h.

**STED imaging.** COS-7 cells were grown on coverslips placed inside 12-well plates. Wells were seeded with ~$1 \times 10^6$ cells in 3 ml cell culture medium. Cells were transfected using FuGene with PIEZO1-HA and PECAM1-SYFP2 per well. After 48 h, cells were fixed and transferred to 12-well plates blocked with 1% BSA (in PBS) (15 min at RT). Primary antibodies against HA (rabbit anti-HA, Cell Signalling Technology, #3724, 1:800) and PECAM1 (mouse anti-PECAM1, 1:200) were diluted in 1% BSA/PBS and added to cells for 1 h at 37 °C. After incubation, cells were washed 3 times (5 min, RT) with PBS. Secondary antibodies against rabbit (anti-rabbit 580, Aberrior, 41367, 1:100) and mouse (anti-mouse STAR red, Aberrior, 52283, 1:100) diluted in PBS were added to the cells and incubated (30 min, RT). Cells were washed 3 times with PBS (5 min, RT) and mounted using ProLong® Gold antifade mountant (ThermoFisher, P36930). STED microscopy was performed using a 100x objective on the STEDYCON 2-colour STED imaging system (Abberior Instruments). Images were exported to FIJI[81] for final processing and assembly. Masks of junctional regions were generated manually using the polygon tool. Images were prepared for particle analysis by using the Gaussian Blur filter (Sigma radius = 0.015 μm in scaled units). Background subtraction was carried out using a sliding paraboloid with rolling ball radius of 3 pixels (0.06 μm). After applying the regional masks, the analysis particles tool was used to identify objects with areas between 0 and 50 pixels ($0–0.02 \mu m^2$) with circularity 0-1. Particle positions were saved in the ROI manager and applied to the raw images to calculate the centres of mass. In a Python script, for the centre of mass of every PIEZO1 particle, we found the distance to the centres of mass of all PECAM1 particles within 0.5 μm in X and Y. From those, we selected the nearest neighbour distance for inclusion in the histogram data. We fitted a parametric distance distribution to this distance histogram with *curve_fit*, a non-linear least-squares method in the *scipy* Python library[82]. The model distribution describes the distance between two 2D Gaussian distributions of localisations. Equivalently, this is the distribution of distances between localisations, where the errors on the localisation coordinates have a 2D Gaussian distribution. The sum of two such distributions was included in the model, which fitted the experimental cell-cell junction data well. The parameter estimates were allowed to vary freely, and results described a prominent proximal distribution of near neighbour distances and a secondary, more distal distribution. The equation was:

$$DD(r) = A_1\left(\frac{r}{\sigma_1^2}\right)\exp\left(-\frac{\mu_1^2 + r^2}{2\sigma_1^2}\right)I_0(r\mu_1/\sigma_1^2) + A_2\left(\frac{r}{\sigma_2^2}\right)\exp\left(-\frac{\mu_2^2 + r^2}{2\sigma_2^2}\right)I_0(r\mu_2/\sigma_2^2)$$

(1)

(adapted from Churchman et al.[57]), where:

- $DD(r)$ is the distribution of distances $r$ from PECAM1 to PIEZO1 localisations
- $\mu_1$ and $\mu_2$ are characteristic distances from PECAM1 to PIEZO1 localisations
- $\sigma_1$ and $\sigma_2$ are the variances of the contributions associated with $\mu_1$ and $\mu_2$
- $A_1$ and $A_2$ are the amplitudes of the contributions associated with $\mu_1$ and $\mu_2$
- $I_0$ is the modified Bessel function of integer order zero.

The parameters of the distance distribution are found by least-squares fitting, which also outputs the covariance matrix for the fitted parameters. From this covariance matrix, confidence intervals for the parameters are calculated and we use 95% confidence intervals as described in Curd et al.[58].

**Red blood cell (RBC) membrane preparation and Western blotting.** RBCs were lysed by incubation for 5 min in a 14× volume of hypotonic solution (46.2 mM NaCl, 0.6 mM HEPES) on ice. Membranes were pelleted by centrifugation at $18,407 \times g$ for 30 min at 4 °C and proteins solubilised using detergent containing lysis buffer (10 mM Tris, pH 7.4, 150 mM NaCl, 0.5 mM EDTA, 0.5% Nonidet P40 substitute, 0.1% glycerol containing protease inhibitor cocktail (Sigma)). Samples were loaded on 7% gels and resolved by electrophoresis. Proteins were transferred to PVDF membranes and labelled overnight with anti-HA (0.01 μg·ml$^{-1}$, Roche clone 3F10). Horse radish peroxidase donkey anti-rat secondary antibodies (1:10000, Jackson ImmunoResearch) and SuperSignal Femto detection reagents (Pierce) were used for visualisation.

**Isolation of endothelium from mouse mesenteric artery.** Endothelium was freshly isolated from second-order branches of mouse mesenteric arteries[38]. Dissected second-order mesenteric arteries were enzymatically digested in dissociation solution containing 126 mM NaCl, 6 mM KCl, 10 glucose, 11 mM HEPES, 1.2 mM MgCl$_2$, 0.05 mM CaCl$_2$ (pH titrated to 7.2) plus 1 mg·ml$^{-1}$ collagenase Type IA (Sigma) for 14 min at 37 °C and then triturated gently to release endothelium on a glass coverslips for recordings on the same day.

**Patch-clamp recording from endothelium.** Recordings were made at room temperature using an Axopatch-200B amplifier equipped with a Digidata 1550 A and pCLAMP 10.6 software (Molecular Devices, Sunnyvale, CA, USA). Endothelium was in a standard bath solution containing 135 mM NaCl, 4 mM KCl, 2 mM CaCl$_2$, 1 mM MgCl$_2$, 10 mM glucose and 10 mM HEPES (titrated to pH 7.4 using NaOH). Membrane potential recordings were made in zero current mode using heat-polished patch pipettes with tip resistances between 3 and 5 MΩ and containing amphotericin B (Sigma-Aldrich) as the perforating agent, added to a pipette solution containing: 145 mM KCl, 1 mM MgCl$_2$, 0.5 mM EGTA and 10 mM HEPES (titrated to pH 7.2 using KOH). Outside-out membrane patch recordings were made in voltage-clamp mode. The tip resistances of recording pipettes were between 12 and 15 MΩ. Currents were sampled at 20 kHz and filtered at 2 kHz. The external solution was standard bath solution and the patch pipette contained: 145 mM KCl, 1 mM MgCl$_2$, 0.5 mM EGTA and 10 mM HEPES (titrated to pH 7.2 using KOH). For application of fluid flow, endothelium or a membrane patch was manoeuvred to the exit of a capillary tube with tip diameter of 350 μm, out of which ionic (bath) solution flowed at 20 μl.s$^{-1}$.

**Infusion of Yoda1 into mice.** Male wild-type (C57BL/6J), 10–14 weeks old, were anaesthetised using an intraperitoneal injection of ketamine hydrochloride (100 mg.kg$^{-1}$) and xylazine hydrochloride (15 mg.kg$^{-1}$). Mice were exsanguinated and then intravenously perfused in situ via portal vein for 10 min at 37 °C with standard bath solution (SBS) containing: 130 mM NaCl, 5 mM KCl, 1.2 mM MgCl$_2$, 8 mM glucose, 10 mM HEPES, 1.5 mM CaCl$_2$, then Yoda1 (3 μM in SBS, 30 min) or DMSO, followed by 4% PFA (10 min) at a flow rate of 1 ml.min$^{-1}$ using a peristaltic pump (Watson-Marlow 505Di). Eyes were harvested for dissection and immunostaining.

**PFA perfusion of PIEZO1[HA] and wild-type mice.** Male, 10–14 weeks old, C57BL/6J wild-type or PIEZO1[HA] mice were anaesthetised under isoflurane (5% induction and 2% maintenance). Mice were perfused via the portal vein by syringe with PBS (10 ml), followed by 4% PFA (20 ml). Eyes were harvested for dissection and immunostaining.

**Dissection and immunostaining of retinas.** Dissection and immunostaining procedures were based on published protocols[83]. Eyes were placed in 4% PFA in PBS for 2 h on ice and washed three times with PBS prior to dissection of retinas. Permeabilisation and blocking of retinas was carried out using staining buffer (PBS pH 6.8, 0.5% Triton, 0.01% Na deoxycholate, 1% BSA, 0.02% NaN$_3$ with (mM) 0.1 CaCl$_2$, 0.1 MgCl$_2$ and 0.1 MnCl$_2$) containing 2% goat serum (Agilent, CA, USA), overnight at 4 °C on an orbital shaker. Primary antibodies against PECAM1 (BD Pharmingen™, 550274, 1:100) and HA (Cell Signalling, mAB3724, 1:100) were diluted in a 1:1 solution of PBS:staining buffer and incubated overnight at 4 °C on an orbital shaker. Retinas were rinsed in PBS with 0.25 % Triton (6×, 15 min) at room temperature. Goat secondary antibodies (Invitrogen A21246 and A11006, 1:200) were diluted in a 1:1 solution of PBS:staining buffer and incubated overnight at 4 °C on an orbital shaker in the dark. Excess antibody was removed by washing with PBS containing 0.25% Triton (6×, 15 min) at room temperature (RT). Retinas were washed in PBS prior to making small quadrantic incisions to allow whole-mounting between a slide and coverslip using ProLong™ Gold (Invitrogen). Imaging was carried out on an LSM710 with 63× oil-immersion objective (Carl Zeiss Ltd.). Images were exported to FIJI for final processing and assembly. Linear adjust of brightness and contrast was applied to the entire image. For quantification of PIEZO1[HA] intensities at the intracellular and junctional regions, masks of the junctions were generated from the corresponding PECAM1 image. The intensity value for each image was normalised to the background. Line profiles were determined using the ImageJ plot profile function. To calculate the coefficient of variance for PECAM1 staining with Yoda1 or DMSO treatment, line profile data

for each image were extracted to OriginPro and normalised to minimum. The coefficient of variance for each trace was plotted.

**FLIM sample preparation**. For cells imaged under static conditions, 35-mm plastic cell culture dishes were plated with ~$3 \times 10^6$ cells in 3 ml cell culture medium. After subculture, cells were transfected, using FuGene, with PIEZO1-mTurquoise. For co-transfections, acceptor-labelled constructs were added. After 48 h, cells were fixed with 4% paraformaldehyde. For flow experiments, cells (~$1 \times 10^6$ in 200 µl medium) were plated onto the ibiTreat µSlide-I$^{0.8}$ Luer (Ibidi) and allowed to attach for 24 h. Transfections were carried out using Lipofectamine™ 3000. The medium in each slide was replaced with the transfection mixture in fresh medium and cells were incubated for 24 h. To culture and stimulate cells under flow conditions we used a system comprising an air pressure pump, pump control software, perfusion set (yellow/green) and fluidics unit (Ibidi). Cells were exposed to shear stress of 10 dyn.cm$^{-2}$ for 24 h. Flow was stopped for a 30 min rest period after which cells were stimulated with flow (10 dyn.cm$^{-2}$, 10 min) and then fixed with 4% PFA.

**FLIM microscopy**. Intensity and FLIM images were obtained on an upright LSM710 (Carl Zeiss) microscope with a 40×/1.0 NA, water-dipping objective or 63×/1.40 Oil (Carl Zeiss). Acceptor intensity images were obtained with excitation at 512 nm using an Argon laser and registered on the Zeiss PMT detectors. Two-photon excitation was provided by Chameleon (Coherent) Ti:Sapphire laser tuned to 800 nm. FLIM emission events were recorded by an external detector (HPM-100, Becker & Hickl) attached to a commercial time-correlated single photon counting electronics module (Becker & Hickl) with a 480/40 (Chroma) emission filter. FLIM images were fitted using in SPCImage (Becker &Hickl). A single component incomplete multi-exponential model was used with a laser repetition time of 12.5 ns. Colour-coded lifetime maps and greyscale intensity images were exported from SPCImage. Acceptor intensity images were processed using FIJI. The histogram intensity weighted mean lifetimes for each image was generated by SPCImage 5.6; values were exported to OriginPro. The peak values were obtained by doing a Gaussian fit.

**Immunostaining**. COS-7 cells (~$1 \times 10^6$.ml$^{-1}$ medium, 30 µl) were plated onto ibiTreat µSlide-VI$^{0.4}$ Luer (Ibidi) and allowed to attach for 24 h. Transfections were carried out using Lipofectamine™ 3000. After 48 h, cells were fixed and blocked with 1 % BSA (in PBS) for 15 min at RT. Primary antibodies against PECAM1 (mouse anti-PECAM1, clone JC70A, Dako1:200) were diluted in 1 % BSA/PBS and added to cells for 1 h at 37 °C. After incubation, cells were washed 3 times (5 min) with PBS. Secondary antibodies mouse (anti-mouse Alexa 594, Jackson Immuno Research, 1:300) diluted in PBS were added to the cells and incubated for 30 min at RT. Cells were washed 3 times with PBS (5 min) and stored in PBS at 4 °C. Samples were imaged on the same day or stored for a maximum of 24 h. Imaging was carried out on LSM710 (Carl Zeiss Ltd.) using a 40×/1.3 oil objective. Images were exported to FIJI for final processing and assembly.

**Patch-clamp on cells overexpressing PIEZO1**. T-REx-293 cells with tetracycline inducible overexpression of human PIEZO1 or T-REx-293 cells with over-expression of mouse PIEZO1 were seeded into a T25 tissue culture flask. After 24 h, transfections were carried out using Lipofectamine™ 3000. Cells were transfected with PECAM1-SYFP2 or SYFP2 (human PECAM1). Expression of PIEZO1 was induced by the addition of tetracycline (100 ng·ml$^{-1}$, 24 h). Cells expressing PIEZO1 with PECAM1-SYFP2 or SYFP2 were detached using trypsin and seeded onto coverslips. Macroscopic transmembrane ionic currents through outside-out patches were recorded using standard patch-clamp technique in voltage-clamp mode. Patch pipettes were fire-polished and had a resistance of 4–7 MΩ when filled with pipette solution. Symmetrical Na$^+$ (K$^+$ / Ca$^{2+}$-free) solution of the following composition: NaCl 140. mM, HEPES 10 mM and EGTA 5 mM (pH 7.4, NaOH), was used for both, pipette and bath solutions, and the currents were recorded at −80 mV. All recordings were made with an Axopatch-200B amplifier (Axon Instruments, Inc., USA) equipped with Digidata 1550B and pClamp 10.6 software (Molecular Devices, USA) at room temperature. 200-ms pressure steps were applied directly to the patch pipette with an interval of 12 s and with an increment of 15 mmHg using High Speed Pressure Clamp HSPC-1 System (ALA Scientific Instruments, USA). Current records were filtered at 2 or 5 kHz and digitally acquired at 5 or 20 kHz.

**Co-immunoprecipitation**. 500 µg of transiently transfected HEK 293 cell lysate (lysis buffer containing 0.5% Nonidet P40 substitute, 0.1% glycerol containing protease inhibitor cocktail (Sigma) and (mM) 10 Tris, 150 NaCl, 0.5 EDTA, at pH 7.4) was incubated with 1 µg anti-HA (Roche, clone 3F10) for at least 4 h at 4 °C prior to extraction overnight using Protein G agarose (Pierce). The beads were washed three times with ice-cold lysis buffer and bound proteins eluted using sample buffer (4× SB: 250 mM Tris pH 6.8, 8% SDS, 40% glycerol, 8% β-mercaptoethanol) and heating at 95 °C. Samples were loaded on 7% gels and resolved by electrophoresis. Proteins were transferred to PVDF membranes and labelled overnight with anti-HA (0.01 µg·ml$^{-1}$, Roche clone 3F10), anti-CDH5 (0.5 µg·ml$^{-1}$, R&D Systems; MAB9381), anti-PECAM1 (1:1000, Dako; clone

JC70A) or anti-β-actin (200 ng·ml$^{-1}$, Santa Cruz). Horseradish peroxidase donkey anti-mouse, anti-rat, anti-goat secondary antibodies (1:10000, Jackson Immuno Research) and SuperSignal Femto detection reagents (Pierce) were used for visualisation. PNGase F (NEB) was used as per manufacturer's instructions.

**GFP-Trap**. T-REx™-293 cells expressing tetracycline-inducible PIEZO1 were transfected in 6 well plates with PECAM1-SYFP2 or PECAM1-Y713F-SYFP2. HEK 293 cells and cells transfected with an empty vector served as controls. Following tetracycline induction, cells were lysed with lysis buffer (10 mM Tris-HCl pH7.5, 150 mM NaCl, 0.5 mM EDTA, 0.5% Nonidet P40 substitute) and centrifuged at 12,000 × *g* for 10 min. Supernatants were quantified, and 400 µg of protein was rotated with GFP-Trap Agarose for 2 h at 4 °C. For input samples, 30 µl of diluted supernatant was removed prior to addition of GFP-Trap Agarose. Following two washes, proteins were eluted with Novex™ Tris-Glycine SDS Sample Buffer (2x) containing 10% (v/v) β-mercaptoethanol at 50 °C for 10 min. Input samples were treated in the same way. Proteins were detected by western blotting (anti-PIEZO1 BEEC4, 1:1000; anti-GFP, Abcam ab1218; 1:5000). BEEC4 is a custom-designed anti-peptide ([C]-DLAKGGTVEYANEKHMLALA) antibody generated in rabbit and affinity-purified by Cambridge Biosciences.

**Halo-Tag pulldowns**. Halo-tagged PIEZO1 was pulled-down using HaloTag® Mammalian Pull-Down Systems (Promega) as per the manufacturer's instructions. Briefly, Griptite™ 293 MSR cells (ThermoFisher) on a 6-well plate were transfected with Halo-PIEZO1 together with human PECAM1. Cells were collected using cold PBS and spun down for 5 min at 500 × *g* at 4 °C to collect the cell pellet. The pellet was frozen at −80 °C for at least 30 min. The pellet was then re-suspended using lysis buffer with 4 µl 50× protease inhibitors from the HaloTag® kit, and homo-genised by passing through a 23 G needle 10 times. The homogenate was centrifuged at 14,000 × *g* for 5 min. The supernatant was rotated with Halo-resin for 45 min at room temperature so that protein complexes containing Halo-tag bound to the resin. Proteins were then eluted with by boiling with 4x Laemmli protein sample buffer (Bio-rad) and detected by western blotting (anti-Halo, 1:1000, Promega, mouse monoclonal and Proteintech PIEZO1 antibody).

**Co-purification**. HEK 293 cells, grown in FreeStyle™ 293 Expression Medium (12338018) to a density of $1 \times 10^6$ cells per ml, were PEI transfected with equal amounts of PIEZO1-GFP and PECAM1 DNA at a PEI to DNA ratio of 3:1[84] and the cells were harvested 72 h post transfection. All procedures following this point were carried out at 4 °C. Cell pellet from 2 L of culture was resuspended in 80 mL buffer (50 mM Tris pH 8, 150 mM NaCl, 1 mM EDTA, 10% glycerol) supplemented with complete protease inhibitor cocktail (Roche), 1 mM AEBSF, 1 mM PMSF. After sonication, the lysate was clarified by centrifugation at 2000 × *g* for 15 min and then supernatant ultra-centrifuged at 100,000 × *g* for 1 h at 4 °C. The membranes were collected and resuspended to a final concentration of 20 mg·ml$^{-1}$ in buffer with protease inhibitors, as above, with the addition of 0.5% lauryl maltose neopentyl glycol (LMNG), 0.1% cholesterol hemisuccinate (CHS) and rotated at 4 °C for 2 h. Unsolubilised material was removed by ultra-centrifugation at 100,000 × *g* for 1 h at 4 °C. The supernatant was then incubated with pre-equilibrated GFP-nanobody coupled Sepharose resin[54] and incubated for 4 h. The resin was loaded onto a column and washed 4 times with 5 column volumes of buffer plus protease inhibitors with decreasing concentrations of LMNG: 0.2, 0.1, 0.01 and 0.005% LMNG. The GST-anti-GFP-nanobody with the protein complex bound was eluted form the GST-resin using 3 column volumes of elution buffer (buffer with 0.005% LMNG and 10 mM reduced L-glutathione) and concentrated to 500 µL by Vivaspin centrifugal filter (300 kDa MWCO). The sample was then loaded onto a Superose 6 increase column equilibrated with 50 mM Tris pH 8, 150 mM NaCl and 0.003 LMNG and fractions corresponding to PIEZO1/PECAM1 complex collected. Bolt™ LDS sample buffer with Bolt™ Sample reducing agent were added to a sample of the fractions (20 µL) and incubated at room temperature (RT) for 20 min before loading onto a Bolt™ 4–12% Bis-Tris gels (all Bolt™ buffer and gels from Invitrogen) and resolving by electrophoresis. The gels were then imaged for GFP fluorescence and Coomassie stained or transferred to PVDF and probed with anti-PECAM1 (1:1000, Dako; clone JC70A) or anti-GST (1:5000, GE Healthcare; 27457701) antibodies. Horse radish peroxidase goat anti-mouse and rabbit anti-goat secondary antibodies (1:5000, Jackson ImmunoResearch) and SuperSignal Femto detection reagents (Pierce) were used for visualisation.

**Cell density experiments**. HUVECs were seeded, 500,000 cells per well, into 6 well plates. After 24 or 48 h of growth cells were fixed with 4% PFA for immu-nostaining or lysed in RIPA buffer supplemented with PMSF, protease inhibitor mixture, and sodium orthovanadate (RIPA Lysis Buffer System, sc24948, Santa Cruz, Dallas, TX) for immunoblotting. All image processing was carried out in FIJI (imageJ). Nuclei counts were obtained from Hoechst images using the imageJ analyse particle function. Junction width was obtained from the αCDH5 immuno-stained images. FIJI line profiles were drawn perpendicular to junctions to generate line profiles. Each profile was fitted with a Gaussian distribution using OriginPro and the full width at half maximum for each peak was measured. F-actin dis-tribution was determined from the phalloidin stained images. ImageJ was used to

draw line profile across each cell perpendicular to the actin filaments. Using OriginPro the coefficient of variance for each line profile was calculated.

**Fura2 calcium measurements**. HUVECs were plated, 60,000 cells per well, into 96 well plates. After 24 h or 48 h of growth cells were incubated for 1 h in Standard Bath Solution (SBS, containing: 130 mM NaCl, 5 mM KCl, 8 mM D-glucose, 10 mM HEPES, 1.2 mM MgCl$_2$, 1.5 mM CaCl$_2$, pH 7.4) supplemented with 2 µM fura-2-AM (F1201, Molecular Probes, Eugene, OR) and 0.01% pluronic acid. Cells were then washed in SBS at room temperature for 30 min. Fura-2. Fluorescence (F) acquisition (excitation 340 and 380 nm; emission 510 nm) was performed on a Flexstation three microplate reader with SoftMax Pro 5.4.5 software (Molecular Devices). After 60 s of recording, Yoda1 (3 µM) or DMSO was injected. Ca$^{2+}$ entry was quantified after normalisation ($\Delta$F340/380 = F340/380(t)-F340/380(t = 0)).

**Calcium switch assay**. HUVECs were transfected with siRNA using Opti-MEM I Reduced Serum Medium (31985070, ThermoFisher Scientific, Waltham, MA) and Lipofectamine 2000 (11668019, ThermoFisher Scientific). For transfection of cells in 6-well plates, a total of 50 nmol of PIEZO1 siRNA (Sigma-Aldrich, GCAA-GUUCGUGCGCGGAUU[DT][DT]) or control siRNA (Dharmacon, L-001810) in 0.1 mL was added to 0.8 mL cell culture medium per well. Medium was changed after 4 h. After 48 h cells were placed in fresh medium for ~30 min before starting the calcium switch assay. Pre-treatment wells were fixed using 4% PFA for 10 min. Calcium depletion and recovery wells were washed with serum-free medium and incubated with serum free HUVEC culture medium containing 3 mM EGTA (Anaspec, ANA84097) for 30 min under culture conditions. Calcium depletion wells were fixed with 4% PFA for 10 min. Recovery wells were washed and covered with HUVEC culture medium containing serum and returned to the culture incubator for 30 min, followed by fixation with 4% PFA, 10 min. Junction thickness and F-actin distributions were calculated as for the cell density experiment.

**Immuno-fluorescence and phalloidin staining of HUVECs**. After fixation, cells were washed with PBS and permeabilised with 0.1% Triton X100 for 10 min. Blocking was carried out with 0.1% BSA for 15 min. Primary antibodies against CDH5 1:300 (Abcam, AB33168) and PECAM1 1:50 (Dako JC70A) were diluted in 0.1%BSA and added to cells overnight at 4 °C. 3 Washes with PBS for 5 min each were followed by incubation with secondary antibodies αrabbit 488, 1:200 (Jackson laboratories, 711-545-152), anti-mouse 647, 1:400 (Thermofisher, A21240) and phalloidin 568 (Cambridge Biosciences, 00044) in 1% BSA. Cells were incubated for 10 min in 1 µg·ml$^{-1}$ Hoechst and washed with PBS.

**Yoda1 treatment of HUVEC monolayers**. HUVECs were seeded, 500,000 cells per well, into 6 well plates. After 48 h of growth cells were treated with either DMSO or 3 µM Yoda1 in serum-free medium and incubated for 30 min under culture conditions. Cells were fixed with 4% PFA for immunostaining. All image processing was carried out in FIJI (imageJ). F-actin distribution was determined from the phalliodin stained images. ImageJ was used to draw line profile across each cell perpendicular to the actin filaments. The coefficient of variance for each line profile was calculated using OriginPro.

**Statistics and reproducibility**. Statistical analysis of fluorescent lifetime values and intensity measurements was carried out using OriginPro. Normality test revealed that at least 1 dataset was not significantly drawn from a normally distributed population at the 0.05 level. Consequently, the Mann–Whitney Test was used to compare if two distributions are significantly different. For data containing 3 different conditions the Kruskal-Wallis Anova was used to check for variance, followed by use of the Mann–Whitney Test to compare data pairs. Each image was treated as an independent replicate, ordering effects were negligible. The number of experiments performed from different cell preparations or animals is defined as n. The study was aimed at deciphering a biological mechanism and so, in the absence of prior knowledge of this mechanism, power calculations were not considered to be applicable. For quantitative data, 3 independent repeats (n) were performed with a minimum of 3 technical repeats each. We selected numbers of independent repeats of experiments based on prior experience of studies of this type. Where multiple pairwise tests were performed on a single dataset, Bonferroni correction was applied. F-test was performed when comparing fitted data. The person performing the patch measurements was blinded to the constructs that had been transfected into the cells. All electrophysiological data were analysed and plotted using pClamp 10.6 and MicroCal Origin 2018 (OriginLab Corporation, USA) Software. Pressure-dependent curves were constructed in Origin and fitted with Boltzmann equation: $y = A2 + (A1 − A2)/(1 + \exp((x − x0)/dx))$, where A1 and A2 are the minima and maxima, x0 the mid-point and dx the slope. For box plot graphs, box = 25%~75%, error bar range within 1.5 interquartile range, median line, □ mean, ◆ outliers (observations that are outside the error bars of the box plot).

**Reporting summary**. Further information on research design is available in the Nature Portfolio Reporting Summary linked to this article.

## Data availability

Uncropped blots are available in Supplementary Fig. 23. The source data behind the graphs in the paper are available in Supplementary Data 1. The source data behind the graphs in the supplementary figures are available in Supplementary Data 2. Supplementary Data 3 contains the table of primers.

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

## Acknowledgements

The study was supported by a Wellcome Investigator Award (110044/Z/15/Z) and British Heart Foundation Programme Grant (RG/17/11/33042) to D.J.B., a British Heart Foundation Intermediate Fellowship to J.S. (FS/17/2/32559), a British Heart Foundation PhD Studentship to H.J.G. (FS/14/22/30734), a BBSRC Grant (BB/S015787/1) to M.P./A.C., a Wellcome ISSF Discipline Hopping Fellowship (204825/Z/16/Z) to A.C. and a British Heart Foundation Project Grant to R.S.B. (PG/19/2/34084). We thank Sally Boxall and Ruth Hughes (Leeds Bio-Imaging Facility) and David Myers for microscopy technical support and Bing Hou and Deborah Linley for experimental data that encouraged elements of the study but which are not included in this article. We thank Lynn McKeown for the PIN-G construct. Figures 7g and 8 were created partly using BioRender.com. For the purpose of Open Access, the authors have applied a CC BY public copyright licence to any Author Accepted Manuscript version arising from this submission.

## Author contributions

E.C.-B., D.J.C. and M.J.L. designed protein constructs. E.C.-B., M.J.L., D.J.C. and H.J.G. cloned protein constructs. E.C.-B. designed and performed microscopy experiments. O.V.P. performed patch-clamp experiments and analysed the data for PIEZO1 over-expression studies. M.J.L., S.T., M.D., C.C.B. designed and performed immunoprecipitation experiments. M.D. performed calcium recordings. D.J.C. performed co-purification studies. S.P.M. performed protein modelling and supervised co-purification studies. A.A. and N.H. genetically engineered mice. T.S.F. bred and maintained genetically engineered mice. J.S. performed patch-clamp studies on endothelium. R.C. advised and assisted on retinal staining experiments. L.L. performed in vivo perfusion and organ harvesting. E.C.-B., O.V.P., A.C., M.P., J.S., M.D. and D.J.B. analysed data. P.D.B. guided statistical analysis of imaging data. E.C.-B. and D.J.B. directed the study, made the figures and wrote the manuscript. D.J.B. conceptualised the study and supervised the project team. D.J.B., J.S. and R.S.B. generated funding.

## Competing interests

The authors declare no competing interests.
