## [Peer Review File · Communications Biology]

Reviewers' comments:

Reviewer #1 (Remarks to the Author):

MS COMMSBIO-22-1039-T, Chuntharpursat et al. Piezo1 channel and CD31 (PECAM1) partnership in endothelial force sensing

Shear stress exerted on a blood vessel wall is a physical phenomenon resulting from the frictional forces generated by the blood flow on the luminal side of the endothelial wall. In contrast to the pressure impulse, which causes a radial stretching of e.g. the arterial wall, the frictional forces of the blood generate a tangential shear stress to the blood flow direction and to the vessel axis, which depends on the instantaneous viscosity of the fluid. It is significantly lower than other mechanical stresses, which means that the sensitivity to shear stress is in a completely different range than other types of mechanosensitivity.

Recent studies have shown that the mechanoregulated plasma membrane ion channel Piezo1 and the mechanocomplex of PECAM1 (CD31), vascular endothelial cadherin (VE-cadherin) and VEGFR2/3 complex, located at cell-cell junctions, are important molecular players in the sensitivity of the endothelium to fluid shear stress. However, the nature of their interactions and whether Piezo1 and/or PECAM1 act as a primary stress sensor have remained controversial until now.

This manuscript intends to describe the localization of Piezo1 at adherens junction, interactions with CD31/PECAM1 and VE-cadherin in the context of cell-cell junction formation, and - in line with previous reports (Nourse et al. 2017, *Semin. Cell Dev. Biol.* 71, 3-12) - the relationship to the EC cytoskeleton. Of particular interest is the authors' hypothesis that CD31 and VE-cadherin are indirectly regulated via Piezo1 by adjusting local calcium levels that influence the remodelling of cell-cell contacts in epithelia. The results are based on optical co-localization studies in mouse endothelial cells, cell-cell contacts of different model cell systems expressing CD31, VE-cadherin and Piezo1, and biochemical experiments. The authors reach their conclusion by combining a variety of different experimental approaches in which Piezo1 is activated either by fluid stress or chemically (Yoda1). However, the manuscript seems to lack a clear experimental rationale and pooled too much data, with often inadequate data analysis and statistics. The introduction does not clearly describe the experimental focus and seems to abruptly move from a general description to a very specific molecular description. I recommend that the manuscript be thoroughly revised, probably involving a selection of key experiments, and clearly indicating the biophysical analysis (see specific comments below).

Specific comments:

1. Line 116 and following: In Figures 1A-C and S1E-G, the authors show patch-clamp recordings of endothelium cell membranes and note that the Piezo1HA channels respond similarly to wild-type channels in the presence of fluid shear stress. This is not made clear enough, in my opinion, as completely different figures and original data are shown for wild-type and mutant channels. What is the origin of the noisy current signal from Piezo1HA in Figure 1A? Why were the data points in Figure 1F connected to each other? A clear side-by-side comparison of the data from both channels would be helpful. In order to compare the conductivities of Piezo1 with the literature, it would also be helpful if the electrolyte concentrations were given in the figure legend. In this context, I also wonder whether the specification and statistics of conductivities to the first decimal is justified for relatively small group numbers of $n < 10$. More clarity is essential, as the first figure sets the stage for the rest of the manuscript.

The authors show that gadolinium blocks the activity of mechanosensitive channels in isolated EC membranes. However, since gadolinium can block the activity of several cation-specific MSCs, it seems important to show that more specific chemical compounds can activate or inhibit Piezo1

channel activity. Another control could be to determine whether Piezo1 acts through its mechanosensitive property and triggers ion channel activities when the cell membrane is stretched by applying negative pressure via the micropipette attached to the membrane.

2. Line 128 - : The authors state that Piezo1 is expressed in retinal venous and microvascular endothelia. Why is a HA intensity signal measured in - CD31 and +CD31 wild-type ECs when the mice do not express HA protein? In my opinion, the dHA signals of about 0.4 afu in figures 1 K,L, Q and R are merely a background. It is also not clear to me what the authors are trying to say with figures 1 I, J, O and P, especially since the HA signals in all tracks between 0 and 20 afu are similar. Why do the HA intensity values in figures I, J, O and P not match those in figures K, L, Q and R?

3. Line 143 - : Co-localization image analyses aim to find indications of possible protein-protein interactions. The latter can be determined by superimposing two images and examining the appearance of the combined color. However, this intuitive method can only work if the intensity values of the two images are similar. The latter does not seem to be the case in Figures 1 I, j, O and P. Based on the information in Materials and Methods (lines 952-974), I am not entirely convinced how to conclude from figures 1G and 1S, U and V that Piezo1 co-localizes with CD31 in the endothelium of retinal vein, unless co-localization is assessed in more quantitative and statistical detail.

4. Line 148 - : Figures 2 A-D show Piezo1 and CD31 in Cos7 cells after heterologous expression by STED microscopy. I wonder how individual Piezo1 and CD31 molecules were identified and why this was done "manually" in selected regions of interest. I have the impression that both proteins are also present in non-cell contact regions. What is meant by selecting particles with a circularity between 0 and 1? Doesn't that mean that any signal with any structure was selected? In general, there are two categories of quantitative approaches to co-localization analysis: intensity-based methods, which calculate global measures using the correlation information of the intensities of two channels ("colors"), and object-based methods, which first segment and identify objects and then consider object distances. Better statistics ($n > 3$) and a clearer step-by-step representation of how the center of mass and the distances between Piezo1 and CD31 were determined in both functional and non-functional regions would strengthen the conclusions drawn from the data. Figure S2A indicates accuracies of distances in the range of 1 nm. How were they determined at a pixel size of 20 nm? What do the authors mean by "parameter uncertainties" in line 172? Since there is no punctate staining of CD31 in Fig. 2B, suggesting that CD31 is very abundant at cell-cell contact sites, I wonder how the histograms shown in Fig. 2E and S2B change when CD31 and/or Piezo1 expression is reduced and when CD31 is mutated at the N- and C-terminus, respectively, to prevent proximity to Piezo1. If the expression level of CD31 or Piezo1 is high, shouldn't the distances between CD31 and Piezo1 automatically be low? In this context, it would be interesting to compare the closest and second closest distances between CD31 and Piezo1 with those of the individual proteins themselves. Another important question is how the effective resolution of the STED micrographs was determined, which is particularly relevant to judge the correctness of Fig. 2E.

5. Line 187 - : While lifetime measurements are suitable to measure the proximity between donor and acceptor dyes, accurate distance measurements are very dependent on the underlying assumptions, especially orientation, motional and concentration effects. It is unclear how the authors approached this question (Fig. 2F-Q).

6. Line 237 - : The authors conclude from patch-clamp experiments that CD31 reduces the mechanical sensitivity of Piezo1 channels when expressed in modified HEK293 cells. Since the error bars are significantly overlapping and to further support this conclusion, it is essential to significantly improve the statistics of the data representing the activation range and to compare the stimulus-response curves with the literature.

7. Line 322 - : The conclusions from Fig. 5, in particular the distance between junctions, do not seem to be justified by the data. In panels Fig. 5 D & E and I & J, the median values appear to be extremely

close. It was also not clear how the intersections were selected for the analyses; what does "manually using a polygon" (line 879) mean?

Minor points:

1. The conclusions drawn from Fig. 1 should be formulated more cautiously.
2. Buffer concentrations should be reported in the same way throughout the manuscript (e.g. "... contains (mM) 135 KCl, 4 KCl ... (lines 923-933)" as opposed to "... 0.1 mM CaCl₂, 0.1 mM MgCl₂ ... (lines 952 - 955)").
3. The legend to figure 2F-H ("FRET/FLIM studies") could be more detailed.

Reviewer #2 (Remarks to the Author):

This manuscript addresses an important topic in the field of vascular mechanobiology: how are linked two of the major mechanosensors, the VE-cadherin junctional complex comprising PECAM, VE-cadherins and VEGFRs on one side and Piezo1 on the other side? The authors suggest there might be a direct link, through a close association of both Piezo1 and PECAM at the endothelial cells junctions.

They engineered a unique mouse strain which expresses a small tag inside Piezo1 sequence, to allow for easy detection and capture from tissues. The function of Piezo1 is not affected by the HA tag, as described by functional patch clamp experiments.

Starting from there, the authors investigate Piezo1-HA localization in blood vessels from the retina, which has the advantage to display three different types of vessels: veinules, arterioles and capillaries. Their first observation, of interest, is that Piezo1-HA is not expressed uniformly. It is also confirmed by a functional experiment whereas Yoda1, a Piezo1 agonist triggers the disassembly of PECAM at the cell-cell junctions.

Recommendation 1: please show complete image of the mouse retina with anti-HA staining.

Recommendation 2: image seems dirty, can you improve the staining, maybe with a different blocking protocol? It is too late now that the study is done but other tags might have improved resolution (Spot-tag for example). The objectives used for the retinas are not specified, as well as the resolution of the images. It would have been nice to get higher resolution images, the use of the term "colocalization" with those images is not adequate. you can only state that Piezo1-HA appears to localise at the junctions.

Recommendation 3: can you also image bigger vessels such as veins and large arteries (aorta for example ??)

Recommendation 4: Effect of Yoda1 might be indirect (huge calcium entry which triggers changes in cell-cell junction structure), it should therefore be explicit to the reader.

Recommendation 5: PECAM should be used instead of CD31.

The authors pursue their study by using COS7 cells expressing both PECAM and Piezo1. They make the argument that those cells are adequate because they do not normally express both proteins. I appreciate the strategy but Piezo CRE recombinase KO mice are available, as well as PECAM KO mice and could have provided endothelial cells without endogenous piezo or PECAM. Indeed, cell-cell junctions in endothelial cells are vastly different from the ones observed in monolayers of COS7 (no

VE-Cad nor VEGFRs are present for example). the other issue is that over expressed PECAM and Piezo1, which is a massive protein complex, will most likely meet and colocalize at the plasma membrane and cell-cell junctions. the image displayed in figure 2A in fact argues against a specific localization: only a tiny portion of the whole cell-cell junction(highlighted) seems to show some specific accumulation of both proteins, most of the time, they seem to be randomly expressed all over the plasma membrane. I am therefore not a big fan of this data set and findings should be toned down or reproduced in endothelial cells.

The authors pursue by performing investigation of direct interaction between the two proteins.

Recommendation 1: Figure 3A is not acceptable in its current form, please provide full blots and images.

Recommendation 2: HEK293 cells are not the best cells to use to perform this experiment. I would recommend to use confluent endothelial cells.

Then they move on a provocative, yet fascinating, idea: PECAM association with Piezo1 would impairs piezo activity. It is well known that Piezo1 quickly inactivates and is therefore almost never fully opened, which makes a lot of sense from a physiological perspective. I like the proposed idea and the data support it. however, I still question the relevance of the model used. The involvement of the N terminal part is adequately supported by a co-IP experiment, although the results are not black or white.

Figure 3I blot is not acceptable in its current form.

Then the authors investigate the role of Piezo1 in the formation of the cell-cell junctions and I have one major comment here: if Piezo1 was necessary, how can you explain that arteries have beautiful cell-cell adherent junctions as you show in figure 1, while they are also lacking Piezo1?

All in all, the authors have identified a very important question and generated intriguing data but this manuscript fails to capture the subject in its whole complexity because the authors tried to do too much, with, in my opinion, inadequate methodology and by jumping too quickly to conclusions. I would recommend them to only focus on some parts of the manuscript, to tone down their claims and to improve a bit the models and data from the mice. There are definitely some aspects of the study which are potentially ground breaking but not in its current form.

Reviewer #3 (Remarks to the Author):

The study addresses a very important goal of finding the connection between two major mechanosensitive endothelial pathways, regulated by PECAM 1 junctional protein and Piezo1 ion channel respectively in order to reconcile the competing hypotheses for endothelial mechanosensation. The study presents multiple lines of evidence using most state-of-the-art approaches to demonstrate physical and functional interactions between the two pathways. The authors also present interesting mechanistic insights into the structural requirements of the interaction. However, the overall picture of how the two pathways interact to confer endothelial shear stress sensitivity remains vague and not clear. There is also no direct evidence that activation of piezo1 by mechanical forces is transduced to the activation of CD31/VE-cadherin junctional complex.

Specific Concerns:

1. The authors show that Piezo1 co-localizes with CD31 in areas of cell-cell contacts, the interaction is verified by FRET. It is also shown that the activation of Piezo1 with its known activator Yoda disrupts CD31 endothelial junctions, but the dispersion of CD31 upon activation of Piezo1 seems to contradict the known effects of laminar flow on the junctions. The more relevant question would be whether Piezo1 activity initiates or enables flow-induced CD31 response.
2. Another interesting observation is that an increase in CD31 expression correlates with and might be causative for reduced sensitivity of Piezo1 to pressure, suggesting a negative feedback but again it is not clear what is the role of this negative feedback in the shear stress-induced mechanosensation. The critical question is whether CD31 is required for piezo1-induced/dependent downstream flow responses.
3. The data about the effects of Piezo1 on the structural integrity of CD31 and VE-cadherin junctions show that there is an effect on the width of the junctions upon depletion/re-introduction of extracellular Ca in the presence vs. the absence of Piezo1. More specifically, they show that Piezo1 KD results in thinner junctions. The structural effect is evident but the functional implications are entirely not clear. In many studies, an increase in junctional width is considered a sign of leakiness. It needs to be tested if this is the case here and if yes, what does it mean for the mechanotransduction.
4. Similarly, a lack of Piezo1 is shown to result in thinner junctional actin, which again is not clear whether it indicates a stronger or weaker cortical actin. An increase in cortical actin is expected to strengthen the junctions but an increased width might indicate weakening of junctions. This needs to be clarified. Also, similarly to the previous comments, a connection with mechano-sensitivity is only implied, not shown

RESPONSES TO REVIEWER #1

Manuscript COMMSBIO-22-1039-T

Reviewer comments are in blue.

Author responses are in black.

Shear stress exerted on a blood vessel wall is a physical phenomenon resulting from the frictional forces generated by the blood flow on the luminal side of the endothelial wall. In contrast to the pressure impulse, which causes a radial stretching of e.g. the arterial wall, the frictional forces of the blood generate a tangential shear stress to the blood flow direction and to the vessel axis, which depends on the instantaneous viscosity of the fluid. It is significantly lower than other mechanical stresses, which means that the sensitivity to shear stress is in a completely different range than other types of mechanosensitivity.

Recent studies have shown that the mechanoregulated plasma membrane ion channel Piezo1 and the mechanocomplex of PECAM1 (CD31), vascular endothelial cadherin (VE-cadherin) and VEGFR2/3 complex, located at cell-cell junctions, are important molecular players in the sensitivity of the endothelium to fluid shear stress. However, the nature of their interactions and whether Piezo1 and/or PECAM1 act as a primary stress sensor have remained controversial until now.

This manuscript intends to describe the localization of Piezo1 at adherens junction, interactions with CD31/PECAM1 and VE-cadherin in the context of cell-cell junction formation, and - in line with previous reports (Nourse et al. 2017, Semin. Cell Dev. Biol. 71, 3–12) - the relationship to the EC cytoskeleton. Of particular interest is the authors' hypothesis that CD31 and VE-cadherin are indirectly regulated via Piezo1 by adjusting local calcium levels that influence the remodelling of cell-cell contacts in epithelia. The results are based on optical co-localization studies in mouse endothelial cells, cell-cell contacts of different model cell systems expressing CD31, VE-cadherin and Piezo1, and biochemical experiments. The authors reach their conclusion by combining a variety of different experimental approaches in which Piezo1 is activated either by fluid stress or chemically (Yoda1).

Thank you for the review of our manuscript and important feedback, which has been very helpful for us. We made changes to address all points. We provide point-by-point responses for all comments. We extensively reorganised the manuscript (text and figures) to communicate the main points better. This has included simplifying the main figures and improving the graphical abstract. Although we provide a track-changes version of the manuscript, it is difficult to follow because of the extensive nature of the changes. Therefore, we highlight specific changes by referring to line numbers in the clean (non-tracked) version. The manuscript is much improved and so we hope you will be happy to consider it further.

However, the manuscript seems to lack a clear experimental rationale and pooled too much data, with often inadequate data analysis and statistics. The introduction does not clearly describe the experimental focus and seems to abruptly move from a general description to a very specific molecular description. I recommend that the manuscript be thoroughly revised, probably involving a selection of key experiments, and clearly indicating the biophysical analysis (see specific comments below).

The rationale was to determine if there is a relationship between the two apparently competing ideas for the sensing of shear stress by endothelium: the PIEZO1 channel, a mediator of mechanically activated calcium ion entry; and PECAM1, a mediator of endothelial cell adhesion and the apex in a triad with VE-cadherin (CDH5) and VEGF receptor 2 (VGFR2). We clarified this in the Abstract (**lines 27-30**) and Introduction (**lines 96-99**). We conclude that there is a relationship, bringing the novel findings of a pool of PIEZO1 at adherens junctions,

PIEZO1 and PECAM1 force sensing mechanisms coming together and intimate cooperation of PIEZO1 and adhesion molecules in the tailoring of junctional structure to mechanical requirement. We reveal molecular details of the interaction, showing that PECAM1 extracellular N-terminus is critical but that there is also contribution of a C-terminal intracellular domain previously found to be tyrosine phosphorylated in response to shear stress.

We revised and expanded the Introduction to describe the focus better and avoid abrupt transition.

We thoroughly revised the entire manuscript as suggested. We restructured the figures, simplified the main figures and strongly reorganised the results text and flow of the experimental focus and rationale. We retained evidence from multiple independent perspectives, which we feel is important for the conclusions, given that all technical approaches have their limitations. In an effort to avoid this clouding the main findings, we moved many of the results to supplementary figures.

We show the PIEZO1^{HA} biophysical investigation and analysis more thoroughly, as described below.

Specific comments:

1. Line 116 and following: In Figures 1A-C and S1E-G, the authors show patch-clamp recordings of endothelium cell membranes and note that the Piezo1HA channels respond similarly to wild-type channels in the presence of fluid shear stress. This is not made clear enough, in my opinion, as completely different figures and original data are shown for wild-type and mutant channels. What is the origin of the noisy current signal from Piezo1HA in Figure 1A? Why were the data points in Figure 1F connected to each other? A clear side-by-side comparison of the data from both channels would be helpful.

We created a new figure (Figure 1), dedicating all of it to this matter. We show more extensive data and data analysis and direct side-by-side comparisons of wildtype and mutant channels.

The noisy current signals for PIEZO1^{HA} and PIEZO1^{WT} activities are due mostly to channel openings and closings with some contribution from background white noise. We dedicated more space to the presentation to make this clearer. We show the full traces on a slow time-base, which creates a diffuse noisy-looking impression. We show parts of the traces on faster time-base to reveal the underlying single channel openings (o1, o2, o3) and closings (c) (Figure 1A, F).

In Figure S1F (number from the original manuscript), we connected the data points together because this shows which 'No flow data point' associates with which 'Flow data point'. They were paired experiments in which the No flow and Flow were applied to the same preparation. We retain this approach. It shows that, in each case, Flow caused depolarisation. We added similar data for PIEZO1^{WT}. In the revised manuscript, these data are in Figure 1E and Figure 1J. To the figure legend, we added (**lines 580-581 and 596-597**): "Data points connected by a line were from the same recording."

In order to compare the conductivities of Piezo1 with the literature, it would also be helpful if the electrolyte concentrations were given in the figure legend.

The electrolyte concentrations have been included in the figure legend of the revised version (Figure 1) as follows (**lines 602-605**):

For the outside-out patch and membrane potential recordings, the external solution consisted of: 135 mM NaCl, 4 mM KCl, 2 mM CaCl₂, 1 mM MgCl₂, 10 mM glucose and 10 mM HEPES

(titrated to pH 7.4 with NaOH). The patch pipette contained: 145 mM KCl, 1 mM MgCl₂, 0.5 mM EGTA and 10 mM HEPES (titrated to pH 7.2 with KOH).

In this context, I also wonder whether the specification and statistics of conductivities to the first decimal is justified for relatively small group numbers of $n < 10$. More clarity is essential, as the first figure sets the stage for the rest of the manuscript.

We retained the first decimal (e.g., 25.9 pS) for consistency and comparison with the original PIEZO paper (Ref. 25: Coste et al 2010 *Science* 330: 55-60).

We dedicate Figure 1 entirely to the topic to help set the stage for what comes.

The authors show that gadolinium blocks the activity of mechanosensitive channels in isolated EC membranes. However, since gadolinium can block the activity of several cation-specific MSCs, it seems important to show that more specific chemical compounds can activate or inhibit Piezo1 channel activity. Another control could be to determine whether Piezo1 acts through its mechanosensitive property and triggers ion channel activities when the cell membrane is stretched by applying negative pressure via the micropipette attached to the membrane.

We previously discussed limitations of gadolinium for PIEZO1 and showed using endothelial-specific knockout methods that these channels are PIEZO1 (Ref. 38: Rode et al 2017 *Nature Commun.* 8:350; Ref. 39: Shi et al 2020 *Cell Rep.* 33:108225). These articles show that PIEZO1 channels are the only resolvable unitary current channel events activated by shear stress in these native endothelial cells (i.e., endothelial cells from second-order mesenteric artery).

The pharmacology available for PIEZO1 channels is limited but we previously showed similar signals activated by the small-molecule agonist of PIEZO1 channels, Yoda1, in mouse mesenteric artery endothelium (Ref. 38: Rode et al 2017 *Nature Commun.* 8:350) or membrane stretch (Ref. 39: Shi et al 2020 *Cell Rep.* 33:108225).

We refer to these matters in the revised manuscript (**Lines 119-140**).

2. Line 128 - : The authors state that Piezo1 is expressed in retinal venous and microvascular endothelia. Why is a HA intensity signal measured in - CD31 and +CD31 wild-type ECs when the mice do not express HA protein?

The no-HA experiments on PIEZO1^{WT} (wild-type) tissue are controls to test for background signals from the anti-HA protocol. We moved such control data to the Supplemental Information (SI) to focus on the main findings and improve clarity (new Figure S4).

In my opinion, the α HA signals of about 0.4 afu in figures 1 K,L, Q and R are merely a background. It is also not clear to me what the authors are trying to say with figures 1 I, J, O and P, especially since the HA signals in all tracks between 0 and 20 afu are similar. Why do the HA intensity values in figures I, J, O and P not match those in figures K, L, Q and R?

The signals of about 0.4 are at or near background. Signals above this are interpreted as arising from PIEZO1^{HA}. They are only detected in retinal vein and capillaries and if CD31 is colocalised (+CD31). To the Results text, we added (**lines 146-149**): "In non-junctional areas without PECAM1 (-PECAM1), the normalised fluorescence signal is close to 0.4 (Figure 2C), which is at or near the background values obtained from PIEZO1^{WT} tissues under similar conditions and the same microscope settings (Figure S4A-I)."

We reorganised this part of the manuscript to make it clearer, showing the primary finding in Figure 2 and supplementary findings and control data in Figure S4.

3. Line 143 - : Co-localization image analyses aim to find indications of possible protein-protein interactions. The latter can be determined by superimposing two images and examining the appearance of the combined color. However, this intuitive method can only work if the intensity values of the two images are similar. The latter does not seem to be the case in Figures 1 I, j, O and P. Based on the information in Materials and Methods (lines 952-974), I am not entirely convinced how to conclude from figures 1G and 1S, U and V that Piezo1 co-localizes with CD31 in the endothelium of retinal vein, unless co-localization is assessed in more quantitative and statistical detail.

The data of Figure 1G, K (numbered from the original manuscript) address colocalisation (new Figure 2A, C). Here, the HA intensity is quantified and compared under identical conditions in the same image for regions with (+CD31) and without (-CD31). The HA intensity is above background only in the +CD31 regions (i.e., at cell-cell junctions) (Figure 2C). This difference is statistically significant and the effect is visually clear. We feel that it is reasonable to conclude from these data that PIEZO1 colocalises with CD31 in situ. Control experimental data are now in Figure S4, showing that signals depend on HA genetic modification.

The data of Figure 1S, U and V (numbered from the original manuscript) are not about colocalisation. They show the effect of the PIEZO1 agonist Yoda1 on CD31 organisation. These data are now in Figures 2D, E and S5.

To improve clarity, we restructured the results section and changed the text, including separate paragraphs for co-localisation (**Line 142**: “PIEZO1^{HA} expression pattern is similar to that of PECAM1 in endothelium”) and effect (**Line 154**: “Pharmacological activation of PIEZO1 disrupts PECAM1 structural organisation”; **Line 166**: “PIEZO1 channel function decreases when the abundance of PECAM1 increases”).

4. Line 148 - : Figures 2 A-D show Piezo1 and CD31 in Cos7 cells after heterologous expression by STED microscopy.

I wonder how individual Piezo1 and CD31 molecules were identified and why this was done “manually” in selected regions of interest.

The density of CD31 at the cell-cell junctions is higher than within the cell, making the junctions distinguishable. We selected these high-density regions with masks drawn manually. We added examples of junctional (new Figure S9E, F) and non-junctional (new Figure S9I, J) masks to show how this was done. An automated approach was used for quantifying puncta but we think the manual approach to selecting regions is the best option.

The data are now in Figure S9 so that we have more space to show examples and explain the experimental approach.

I have the impression that both proteins are also present in non-cell contact regions.

Both protein are also present in non-cell contact (non-junctional) regions. The new Figure S9I-K shows this in more detail. Further data for non-junctional regions are in the new Figures 4D-L, S12C, 6A-E and S15C.

What is meant by selecting particles with a circularity between 0 and 1? Doesn't that mean that any signal with any structure was selected?

No size or shape constraints were necessary for particle detection by the ImageJ particle finder algorithm, hence the circularity settings of 0 and 1. Any signal with any structure was selected as long as it was above background and within 0-50 pixels (0-0.02 μm^2). A statement to this effect is in the “STED imaging” section of the Methods (**lines 902-904**). The new Figure S9E, F, I, J panels were added to visualise it.

In general, there are two categories of quantitative approaches to co-localization analysis: intensity-based methods, which calculate global measures using the correlation information of the intensities of two channels (“colors”), and object-based methods, which first segment and identify objects and then consider object distances.

We revised the STED imaging text (**lines 225-231**) and associated figure (new Figure S9) to clarify that our approach here was to analyse the distribution of distances between PIEZO1^{HA} and CD31. We did not use a correlation of intensities method.

Better statistics ($n > 3$) and a clearer step-by-step representation of how the center of mass and the distances between Piezo1 and CD31 were determined in both functional and non-functional regions would strengthen the conclusions drawn from the data.

The n in this case was the number of completely independent experimental repeats but the depth of analysis was much greater in each repeat. These are long challenging experiments and we feel that the approach is reasonable, normal for high quality studies in the field and sufficient for our conclusions. The histogram was aggregated from 16 fields of view obtained over 3 independent repeats which is the standard number of independent repeats used for statistical analysis, though no statistical comparisons are made for these data. Nearest neighbour distances were calculated for a total of 598 PIEZO1 particles and 993 CD31 PECAM1 particles in the junctional regions and 1402 PIEZO1 particles and 2597 CD31 particles in the non-junctional regions.

We have expanded the presentation. The new Figure S9 includes the masks used to determine junctional and non-junctional regions and the particles within each of these regions as determined by our analysis.

Figure S2A indicates accuracies of distances in the range of 1 nm. How were they determined at a pixel size of 20 nm? What do the authors mean by “parameter uncertainties” in line 172?

These parameters are from model fitting of continuously variable parameters of the model distance distribution $DD(r)$ to the distance histogram. The centre of a Gaussian distribution over several pixels could be found at any point within a pixel, not only at a corner or the centre, for instance. Therefore, the model parameters are not quantised to multiples of a single pixel. Furthermore, uncertainties in these parameter estimates are given, calculated from the covariance matrix of the fitted model, as described in Curd et al. (Ref. 58: Nanoscale Pattern Extraction from Relative Positions of Sparse 3D Localizations. *Nano Lett* 21, 1213-1220, doi:10.1021/acs.nanolett.0c03332 (2021)).

Additionally, the positions of the centres of mass found for the fluorescent signals are calculated using the standard ImageJ algorithm to below 1 nm precision.

In the related field of localisation microscopy, position estimates for single molecules are routinely also found by template matching or model fitting at $\sim 1/100^{\text{th}}$ of the pixel size, with position uncertainty estimates of $\sim 10^{\text{th}}$ of the pixel size.

Therefore, model fitting results for characteristic distances at sub-1 nm precision are reasonable at 20 nm pixel size, especially when also including the estimate of their uncertainty.

We have included symbols from the equation in Methods, STED imaging, in the new Figure S9H.

The parameters of the distance distribution (Methods) are found by least-squares fitting, which also outputs the covariance matrix for the fitted parameters. From this covariance matrix, confidence intervals for the parameters are calculated, as described in Curd et al. (Ref. 58, as above). We use 95% confidence intervals calculated in this way (e.g., as given in the new Figure S9H).

We clarified in the Methods by including the following (**lines 925-928**): “The parameters of the distance distribution are found by least-squares fitting, which also outputs the covariance matrix for the fitted parameters. From this covariance matrix, confidence intervals for the parameters are calculated and we use 95% confidence intervals as described in Curd et al (ref. 58).”

Since there is no punctate staining of CD31 in Fig. 2B, suggesting that CD31 is very abundant at cell-cell contact sites, I wonder how the histograms shown in Fig. 2E and S2B change when CD31 and/or Piezo1 expression is reduced and when CD31 is mutated at the N- and C-terminus, respectively, to prevent proximity to Piezo1.

To investigate proximity at low expression level and with CD31 mutation, we used the more powerful approach of FRET/FLIM (new Figures 4, S10, S12, S15 and S17). We show, for example, that the proximity at cell-cell contact points (junctional sites) is as close as for 2 PIEZO1s (new Figure S10C *cf* new Figure 4C), not replicated by another protein VEGFR2 (new Figure 5D) and disrupted by some (e.g., Y713F) but not all C-terminal CD31 mutations (new Figure 4E). We show that proximity of CD31 N-terminus to PIEZO1 is specific to the junctional sites (new Figure 4K, L).

If the expression level of CD31 or Piezo1 is high, shouldn't the distances between CD31 and Piezo1 automatically be low?

Our CD31 mutant and VEGFR2 data (indicated above) argue against this; i.e., the effects are not non-specific.

In this context, it would be interesting to compare the closest and second closest distances between CD31 and Piezo1 with those the individual proteins themselves.

We addressed this using FRET/FLIM and the proximity of 2 PIEZO1s, as indicated above. These data suggest that the proximity of CD31 to PIEZO1 is similar to that of 2 PIEZO1s. This is consistent with published work suggesting that PIEZO1 binds cell adhesion molecules directly, which we mention in the text (Ref. 73: Wang et al, *Cell Rep* 38, 110342, doi:10.1016/j.celrep.2022.110342 (2022)).

Another important question is how the effective resolution of the STED micrographs was determined, which is particularly relevant to judge the correctness of Fig. 2E.

As described above, the centres of mass are found to a small fraction of the pixel size using the ImageJ algorithm indicated in the text. The proximity determined by STED provided us with the incentive to progress to FRET/FLIM and the extensive data sets we show with this technique.

5. Line 187 - : While lifetime measurements are suitable to measure the proximity between donor and acceptor dyes, accurate distance measurements are very dependent on the underlying assumptions, especially orientation, motional and concentration effects. It is unclear how the authors approached this question (Fig. 2F-Q).

It is correct that with FRET/FLIM we measure the proximity between the tags that need to be ~2.6 nm apart to allow energy transfer, which is on the order of the average protein radius. For flexibility in orientation and mobility, we inserted a linker between the target protein and fluorescent tag for both constructs, as now specified in the Results text (**lines 239-240**: “We inserted a linker between the target protein and fluorescent tag for both constructs”).

To minimise risks from high concentrations of protein, we optimised transfections and microscopy for low expression levels while retaining confidence in detection, measuring lifetime values over a range of donor-acceptor stoichiometry. We included controls such as the PIEZO1-PIEZO1, CD31 mutation and VEGFR2 studies as described above. In addition we validated our results by copurification and coimmunoprecipitation of the partner proteins.

We include a paragraph in the Discussion addressing this matter (**lines 479-494**): “Some of our conclusions are based on expression of tagged proteins in model cell systems, including non-endothelial cells so that we could utilise their technical advantages and reconstitute effects on null backgrounds for PIEZO1, PECAM1 and CDH5. Such technical approaches confer accuracy and reliability, are flexible, can be robustly tested and controlled and enable the relatively simple testing of the roles of specific amino acid residues through expression of mutants. However, there are potential limitations. The host cell type may not handle the exogenous proteins in the same way as the native cell. The experiments may involve overexpression (i.e., abnormally high protein expression that does not occur physiologically and which may cause effects that are not physiological). We took care to minimise the risk of abnormal expression, using transfection efficiencies that were just sufficient to enable detection of tagged constructs in natural cellular patterns. We also used multiple methods and cell types to validate our conclusions as each system has different limitations. The arising data provide a framework for designing future studies of such mechanisms in vivo, for example through the engineering of mice containing mutant or fluorescently-tagged endogenous proteins that can be studied in the native context of endothelial cells.”

6. Line 237 - : The authors conclude from patch-clamp experiments that CD31 reduces the mechanical sensitivity of Piezo1 channels when expressed in modified HEK293 cells. Since the error bars are significantly overlapping and to further support this conclusion, it is essential to significantly improve the statistics of the data representing the activation range and to compare the stimulus-response curves with the literature.

The error bars are standard deviation (SD), so overlap does not indicate lack of statistical significance. We used a blinded unbiased experimental approach that did not select data based on any characteristics. All data are included and the full variability is shown by the SD and the display of all original data points. We analysed the amplitude at 3 pressure pulses (new Figure 3I, J) and determined the position of the stimulus-response curve by curve-fitting (new Figure 3K). In both cases, there is statistical significance (new Figure 3, legend).

The average pressures for half-activation (P_{50S}) of human PIEZO1 and mouse PIEZO1 (with SYFP2 but not CD31) were 68 mmHg (new Figure S6C) and 49 mmHg (new Figure S7) respectively. We are not aware of comparable data in the literature in the presence of SYFP2 but note the similarity of our stimulus-response data to those reported previously for positive pressure steps applied to outside-out patches. We revised the text as follows (**lines 179-183**): “We used excised outside-out patches for data collection from the surface membrane and applied positive pressure steps to mechanically activate the channels, as previously reported (Ref. 52). As expected (Ref. 52), cells expressing PIEZO1 show prominent mechanically activated macroscopic currents (Figure 3H).” Reference 52 (Lewis & Grandl 2015) systematically compared PIEZO1 properties for different patch configurations and pressure-pulse protocols; it was noted that “For outside-out patches, much larger pressures were required for activation than other configurations (positive pressure: $P_{50} = +70.0 \pm 9.6$ mmHg)”

7. Lin3 322 - : The conclusions from Fig. 5, in particular the distance between junctions, do not seem to be justified by the data.

We quantified the junction width from images of the type represented in Figure 4A and B using a line profile as shown in Figure 4C with quantification from 3 independent repeats, 9 fields of view and ~90 junctions for each condition. We also quantified the number of actin fibres per cell from images of the type represented in Figure 4 F and G using a line profile as shown in Figure 4H with quantification from 3 independent repeats, 9 fields of view and ~90 cells for each condition. We did not aim to quantify the distance between junctions or formulate any conclusions regarding the distance between junctions.

In panels Fig. 5 D & E and I & J, the median values appear to be extremely close.

The actual median and mean values are inserted here for clarity. In some instances, they are close (and not different) but in other instances there are quite large difference (25-50%) and they are statistically significant, as clarified below for each data set.

Fig. 5D Ctrlsi mean: 2.76799 μm

Fig. 5D Ctrlsi median: 2.22302 μm

Fig. 5D PIEZO1si mean: 2.74395 μm

Fig. 5D PIEZO1si median: 2.24936 μm

The Ctrlsi and PIEZO1si values are close and there is no statistically significant difference.

Fig. 5E Ctrlsi mean: 2.30383 μm

Fig. 5E Ctrlsi median: 1.97278 μm

Fig. 5E PIEZO1si mean: 1.75102 μm

Fig. 5E PIEZO1si median: 1.42495 μm

The PIEZO1si values are about 25% less and there is statistically significant difference.

Fig. 5I Ctrlsi mean: 0.23401 peaks/ μm

Fig. 5I Ctrlsi median: 0.22029 peaks/ μm

Fig. 5I PIEZO1si mean: 0.22374 peaks/ μm

Fig. 5I PIEZO1si median: 0.22734 peaks/ μm

The Ctrlsi and PIEZO1si values are close and there is no statistically significant difference.

Fig. 5J Ctrlsi mean: 0.15896 peaks/ μm

Fig. 5J Ctrlsi median: 0.15381 peaks/ μm

Fig. 5J PIEZO1si mean: 0.1197 peaks/ μm

Fig. 5J PIEZO1si median: 0.1104 peaks/ μm

The PIEZO1si values are about 30% less and there is statistically significant difference.

Fig. 5K Ctrlsi mean: 0.25164 peaks/ μm

Fig. 5K Ctrlsi median: 0.27077 peaks/ μm

Fig. 5K PIEZO1si mean: 0.12244 peaks/ μm

Fig. 5K PIEZO1si median: 0.11963 peaks/ μm

The PIEZO1si values are about 50% less and there is statistically significant difference.

The figure labels above are from the original manuscript. Fig. 5D, E is now Fig. 7D, E and Fig. 5I-K is now Fig. S21D-F.

It was also not clear how the intersections were selected for the analyses; what does "manually using a polygon" (line 879) mean?

Methods described in line 879 (now **line 899**) are for Figure 2A-E (now Figure S9). The density of CD31 at the cell-cell junctions is higher than within the cell, making the junctions distinguishable. We selected these high-density regions with masks drawn manually. To improve clarity, we added examples of junctional masks to show how this was done (new Figure S9E).

In Figure 5 (now Figure 7), junctions were identified by the enrichment of VE-Cadherin (green). Approximately 10 junctions per field of view from were picked at random and analysed. We had 9 fields of view per condition from 3 independent experiments. The junction width was determined using line profiles as illustrated in Figure 5C (now Figure 7C).

Minor points:

1. The conclusions drawn from Fig. 1 should be formulated more cautiously.

We reorganised, enhanced and rewrote most of the text and hope it is more satisfactory.

2. Buffer concentrations should be reported in the same way throughout the manuscript (e.g. "... contains (mM) 135 KCl, 4 KCl ... (lines 923-933)" as opposed to "... 0.1 mM CaCl₂, 0.1 mM MgCl₂ ... (lines 952 - 955)".

We revised for consistent formatting.

The legend to figure 2F–H ("FRET/FLIM studies") could be more detailed.

This is the new Figure 4A-C and the figure legend was revised as follows (**lines 664-676**):

(A) PIEZO1-mTurquoise2 alone. The image on the left is an intensity image in which white is high intensity. The image on the right is a lifetime image calibrated to the rainbow scale indicated at the top right corner (3.5 to 4.2 ns). The white arrows point to a region of cell-cell contact and the orange arrowheads nucleus (N) and endoplasmic reticulum (ER). The graph on the right is the lifetime distribution with the limits of the rainbow scale indicated by vertical lines.

(B) Similar to **(A)** except PIEZO1-mTurquoise2 with PECAM1-SYFP2 (+PECAM1). **(A, B)** Scale bars, 50 μ m.

(C) Box plot presentation of summary data of the type shown in **(A, B)**, showing peak lifetime for the entire cell (Whole cell). n = 4 independent experiment repeats for PIEZO1-mTurquoise2 alone (-PECAM1, black) and PIEZO1-mTurquoise2 plus PECAM1-SYFP2 cells (+PECAM1, red). *** $P = 3.16 \times 10^{-5}$. The superimposed data points are average lifetimes for different images, -PECAM1 (N = 11) and +PECAM1 (N = 13)."

RESPONSES TO REVIEWER #2

Manuscript COMMSBIO-22-1039-T

Reviewer comments are in blue.

Author responses are in black.

This manuscript addresses an important topic in the field of vascular mechanobiology: how are linked two of the major mechanosensors, the VE-cadherin junctional complex comprising PECAM, VE-cadherins and VEGFRs on one side and Piezo1 on the other side? The authors suggest there might be a direct link, through a close association of both Piezo1 and PECAM at the endothelial cells junctions.

They engineered a unique mouse strain which expresses a small tag inside Piezo1 sequence, to allow for easy detection and capture from tissues. The function of Piezo1 is not affected by the HA tag, as described by functional patch clamp experiments.

Starting from there, the authors investigate Piezo1-HA localization in blood vessels from the retina, which has the advantage to display three different types of vessels: veinules, arterioles and capillaries. Their first observation, of interest, is that Piezo1-HA is not expressed uniformly. It is also confirmed by a functional experiment whereas Yoda1, a Piezo1 agonist triggers the disassembly of PECAM at the cell-cell junctions.

Thank you for the review of our manuscript and important feedback, which has been very helpful for us. We made changes to address all points. We provide point-by-point responses for all comments. We extensively reorganised the manuscript (text and figures) to communicate the main points better. This has included simplifying the main figures and improving the graphical abstract. Although we provide a track-changes version of the manuscript, it is difficult to follow because of the extensive nature of the changes. Therefore, we highlight specific changes by referring to line numbers in the clean (non-tracked) version. The manuscript is much improved and so we hope you will be happy to consider it further.

Recommendation 1: please show complete image of the mouse retina with anti-HA staining.

We inserted a complete image of retina (new Figure S3).

Recommendation 2: image seems dirty, can you improve the staining, maybe with a different blocking protocol? It is too late now that the study is done but other tags might have improved resolution (Spot-tag for example). The objectives used for the retinas are not specified, as well as the resolution of the images. It would have been nice to get higher resolution images, the use of the term "colocalization" with those images is not adequate. you can only state that Piezo1-HA appears to localise at the junctions.

We optimised the protocols before conducting these experiments and made side-by-side comparisons of control (no HA, wild-type [WT]) and test (HA) images. The expression level of the PIEZO1 is relatively low, making it difficult to avoid background signals. There are nevertheless similar background speckles in both (e.g., new Figure 2A α HA PIEZO1^{HA} and new Figure S4A α HA PIEZO1^{WT}). Such artefacts provide reassurance that comparable conditions and microscope settings were used for both groups. The artefacts do not adversely affect the analysis or conclusions.

We added (**lines 993-994**) "Imaging was carried out on a LSM710 with 63x oil-immersion objective".

Recommendation 3: can you also image bigger vessels such as veins and large arteries (aorta for example)?

We imaged thoracic aorta and portal vein. Consistent with the retinal studies shown in the manuscript (e.g., new Figure 2), there is more PIEZO1 in vein (figure inserted here). We did not add these data to the revised manuscript because other reviewers felt that we should reduce overall what we show and reduce complexity. We nevertheless hope to publish these data in a separate article about PIEZO1 in portal vein function.

Recommendation 4: Effect of Yoda1 might be indirect (huge calcium entry which triggers changes in cell-cell junction structure), it should therefore be explicit to the reader.

We revised the relevant section of the Results and added data as follows:

Lines 154-164: “Pharmacological activation of PIEZO1 disrupts PECAM1 structural organisation To determine if PIEZO1 has functional significance for in situ PECAM1, we infused PIEZO1 channel small-molecule agonist Yoda1⁵¹ for 30 min in vivo, using exsanguinated mice to minimise problems due to potential Yoda1 instability and plasma protein binding. Mice were then perfusion-fixed and retinal vasculature was stained. We showed previously that the effects of Yoda1 on mouse endothelial cells are abolished by endothelial-specific PIEZO1 deletion³⁸. In the retina, Yoda1 disorders the pattern of PECAM1, notably in vein (Figure 2D, E, Figure S5A) but not artery (Figure S5B-D). The venous specificity aligns with the venous localisation (Figure 2A, C), consistent with the idea that Yoda1 acts via PIEZO1. The data suggest that PIEZO1 is capable of regulating the organisation of PECAM1 at endothelial cell-cell junctions.”

Lines 342-349: “Pharmacological activation of PIEZO1 causes radial actin collapse External force was not applied in the studies of Figure 7 and Figure S21 and so we assume PIEZO1 was activated only by physiological forces inherent to the cells and their substrate. Application of the small-molecule agonist of PIEZO1, Yoda1, near its concentration for 50% effect (3 μ M)⁶⁶ causes substantial intracellular Ca²⁺ elevation above basal levels of such cells,

suggesting strong additional PIEZO1 activation (Figure S19A, B). This concentration of Yoda1 strikingly disorganises F-actin structure (Figure S22). Something similar may have occurred when Yoda1 was infused in situ (Figure 2D, E).”

Recommendation 5: PECAM should be used instead of CD31.

We changed throughout to PECAM1 instead of CD31 in the text and figures. We similarly adopted standard names for other proteins such as VE-cadherin (CDH5). We specify other commonly used names.

The authors pursue their study by using COS7 cells expressing both PECAM and Piezo1. They make the argument that those cells are adequate because they do not normally express both proteins. I appreciate the strategy but Piezo CRE recombinase KO mice are available, as well as PECAM KO mice and could have provided endothelial cells without endogenous piezo or PECAM. Indeed, cell-cell junctions in endothelial cells are vastly different from the ones observed in monolayers of COS7 (no VE-Cad nor VEGFRs are present for exemple). the other issue is that over expressed PECAM and Piezo1, which is a massive protein complex, will most likely meet and colocalize at the plasma membrane and cell-cell junctions. the image displayed in figure 2A in fact argues against a specific localization: only a tiny portion of the whole cell-cell junction(highlighted) seems to show some specific accumulation of both proteins, most of the time, they seem to be randomly expressed all over the plasma membrane. I am therefore not a big fan of this data set and findings should be toned down or reproduced in endothelial cells.

PIEZO1 and PECAM1 are key proteins in endothelial cells, helping to make the cells endothelial. Deleting these proteins profoundly alters the endothelial cells, affecting their ability to respond to shear stress and form cell-cell junctions. Such cells may not be an improvement on COS7 cells.

To understand molecular details of PIEZO1-PECAM1 interactions in native endothelium, we need to first determine principles in reconstituted systems, as we show for COS7 cells. We can then design in situ strategies based on predictions from such approaches and thereby seek to manipulate the interactions and observe the consequences.

We include a paragraph in the Discussion addressing this matter (**lines 479-494**): “Some of our conclusions are based on expression of tagged proteins in model cell systems, including non-endothelial cells so that we could utilise their technical advantages and reconstitute effects on null backgrounds for PIEZO1, PECAM1 and CDH5. Such technical approaches confer accuracy and reliability, are flexible, can be robustly tested and controlled and enable the relatively simple testing of the roles of specific amino acid residues through expression of mutants. However, there are potential limitations. The host cell type may not handle the exogenous proteins in the same way as the native cell. The experiments may involve overexpression (i.e., abnormally high protein expression that does not occur physiologically and which may cause effects that are not physiological). We took care to minimise the risk of abnormal expression, using transfection efficiencies that were just sufficient to enable detection of tagged constructs in natural cellular patterns. We also used multiple methods and cell types to validate our conclusions as each system has different limitations. The arising data provide a framework for designing future studies of such mechanisms in vivo, for example through the engineering of mice containing mutant or fluorescently-tagged endogenous proteins that can be studied in the native context of endothelial cells.”

The authors pursue by performing investigation of direct interaction between the two proteins.

Recommendation 1: Figure 3A is not acceptable in its current form, please provide full blots and images.

All original uncropped blots are in the data source file.

Recommendation 2: HEK293 cells are not the best cells to use to perform this experiment. I would recommend to use confluent endothelial cells.

Please see our response above to “The authors pursue their study by using ...”.

Then they move on a provocative, yet fascinating, idea: PECAM association with Piezo1 would impair piezo activity. It is well known that Piezo1 quickly inactivates and is therefore almost never fully opened, which makes a lot of sense from a physiological perspective. I like the proposed idea and the data support it. However, I still question the relevance of the model used. The involvement of the N terminal part is adequately supported by a co-IP experiment, although the results are not black or white.

We understand that no response is required here, beyond the points we make above.

Figure 3I blot is not acceptable in its current form.

We apologise for the error, which has been corrected.

Then the authors investigate the role of Piezo1 in the formation of the cell-cell junctions and I have one major comment here: if Piezo1 was necessary, how can you explain that arteries have beautiful cell-cell adherent junctions as you show in figure 1, while they are also lacking Piezo1?

We changed the text to read: **(lines 149-151)** “PIEZO1^{HA} is not detected in retinal artery, although PIEZO1 may be in these arteries (Ref. 50) but below the threshold for detection in our assay (Figure S4D-I)”; and **(lines 160-162)** “Yoda1 disorders the pattern of PECAM1, notably in vein (Figure 2D, E, Figure S5A) but not artery (Figure S5B-D). The venous specificity aligns with the venous localisation (Figure 2A, C), consistent with the idea that Yoda1 acts via PIEZO1.”

We had included a Discussion paragraph about this matter, but deleted it in revision to improve the focus on the manuscript. This paragraph is as follows and may go some way to addressing your valid question: “A finding from our HA-tagged PIEZO1 mouse is that there is more PIEZO1 in retinal vein than retinal artery. Such venous preference was independently validated by the PIEZO1 agonist (Yoda1) infusion experiments. In support of our venous localisation observations, scRNAseq analysis has suggested higher venous than arterial PIEZO1 mRNA in endothelial cells from mouse. Vascular structure, cell biology and fluid dynamics are known to be diverse, so it can be expected that further differential roles of PIEZO1 will emerge. The preference for venous function is consistent with human genetic studies, which have suggested particular importance of PIEZO1 in varicose vein pathology. Genetic studies have also pointed to greater importance of PIEZO1 in lymphatic vessels, suggesting roles of PIEZO1 in tissue drainage mechanisms. Nevertheless, the specific significance of preferential PIEZO1 expression in veins over arteries is unknown. It remains to be determined if PIEZO1 is simply less but still sufficient in artery or if another ion channel such as PIEZO2 takes the place of PIEZO1 in adherens junctions of retinal arteries. PIEZO1 channels are not completely absent from endothelium of arteries. We previously reported mild but significant responses to Yoda1 in aorta. We also detected PIEZO1 channel activity in endothelium of mesenteric artery and indeed PIEZO1^{HA} channel activity in mesenteric artery endothelial cells in this study. The presence of PIEZO1^{HA} in retinal microvasculature and activity of PIEZO1 channels shown electrophysiologically in retinal vasculature adds to an emerging picture of PIEZO1 as a mechanical sensor of capillaries.”

All in all, the authors have identified a very important question and generated intriguing data but this manuscript fails to capture the subject in its whole complexity because the authors tried to do too much, with, in my opinion, inadequate methodology and by jumping too quickly to conclusions. I would recommend them to only focus on some parts of the manuscript, to tone down their claims and to improve a bit the models and data from the mice. There are definitely some aspects of the study which are potentially ground breaking but not in its current form.

We thoroughly revised the entire manuscript. We restructured the figures, simplified the main figures and strongly reorganised the results text and flow of the experimental focus and rationale. We retained evidence from multiple independent perspectives, which we feel is important for the conclusions, given that all technical approaches have their limitations.

We improved the central message of the work through many changes, including to the Discussion. We illustrate the concepts better through the new Figure 7 and new Figure 8 and associated text.

RESPONSES TO REVIEWER #3

Manuscript COMMSBIO-22-1039-T

Reviewer comments are in blue.

Author responses are in black.

The study addresses a very important goal of finding the connection between two major mechanosensitive endothelial pathways, regulated by PECAM 1 junctional protein and Piezo1 ion channel respectively in order to reconcile the competing hypotheses for endothelial mechanosensation. The study presents multiple lines of evidence using most state-of-the-art approaches to demonstrate physical and functional interactions between the two pathways. The authors also present interesting mechanistic insights into the structural requirements of the interaction. However, the overall picture of how the two pathways interact to confer endothelial shear stress sensitivity remains vague and not clear. There is also no direct evidence that activation of piezo1 by mechanical forces is transduced to the activation of CD31/VE-cadherin junctional complex.

Thank you for the review of our manuscript and important feedback, which has been very helpful for us. We made changes to address all points. We provide point-by-point responses for all comments. We extensively reorganised the manuscript (text and figures) to communicate the main points better. This has included simplifying the main figures and improving the graphical abstract. Although we provide a track-changes version of the manuscript, it is difficult to follow because of the extensive nature of the changes. Therefore, we highlight specific changes by referring to line numbers in the clean (non-tracked) version. The manuscript is much improved and so we hope you will be happy to consider it further.

We bring the novel findings of a pool of PIEZO1 at adherens junctions, PIEZO1 and PECAM1 force sensing mechanisms coming together and intimate cooperation of PIEZO1 and adhesion molecules in the tailoring of junctional structure to mechanical requirement. We reveal molecular details of the interaction, showing that PECAM1 extracellular N-terminus is critical but also show contribution of a C-terminal intracellular domain that was previously found to be tyrosine phosphorylated in response to shear stress. Nevertheless, we recognise that we are on a journey to understanding this complex situation. There are unanswered questions and the physiological nature of the relationship between these systems in response to fluid flow remains to be solved.

We thoroughly revised the entire manuscript to improve clarity. We restructured the figures, simplified the main figures and strongly reorganised the results text and flow of the experimental focus and rationale. We retained evidence from multiple independent perspectives, which we feel is important for the conclusions, given that all technical approaches have their limitations.

We added further discussion to better recognise the key points you raise below about the response to fluid flow in a physiological setting.

We clarified the key steps forward, improving the description of our model in the new Figure 7G and new Figure 8.

Specific Concerns:

1. The authors show that Piezo1 co-localizes with CD31 in areas of cell-cell contacts, the interaction is verified by FRET. It is also shown that the activation of Piezo1 with its known activator Yoda disrupts CD31 endothelial junctions, but the dispersion of CD31 upon activation of Piezo1 seems to contradict the known effects of laminar flow on the junctions.

We added:

Lines 342-349:

“Pharmacological activation of PIEZO1 causes radial actin collapse External force was not applied in the studies of Figure 7 and Figure S21 and so we assume PIEZO1 was activated only by physiological forces inherent to the cells and their substrate. Application of the small-molecule agonist of PIEZO1, Yoda1, near its concentration for 50% effect (3 μ M) (Ref. 62) causes substantial intracellular Ca^{2+} elevation above basal levels of such cells, suggesting strong additional PIEZO1 activation (Figure S19A, B). This concentration of Yoda1 strikingly disorganises F-actin structure (Figure S22). Something similar may have occurred when Yoda1 was infused in situ (Figure 2D, E).”

Lines 510-516:

“We show that pharmacological activation of PIEZO1 (by the agonist Yoda1) disrupts PECAM1’s structural organisation and causes the collapse of radial actin. Our data suggest that Yoda1 can activate PIEZO1 substantially above its physiological activation levels, generating effects that are not necessarily physiologically relevant. In future studies, titrating Yoda1’s concentration down may lead to better mimicry of PIEZO1 activation by physiological force, although it may also be necessary to apply the Yoda1 in a localised and directional manner, as occurs with forces such as shear stress and membrane tension.”

We clarified our model for the physiology in the new Figure 7G and new Figure 8.

The more relevant question would be whether Piezo1 activity initiates or enables flow-induced CD31 response.

Our focus has been structural and molecular mechanisms, which we feel is necessary at this stage. We recognise that we are still some way from revealing what happens in a physiological setting in response to flow. We worked to improve the physiological aspects of the data (e.g., new Figure 7) and added discussion to better contextualize our progress (e.g., **lines 457-477**):

“We reveal new molecular knowledge and bring apparently competing ideas closer together but there remain major questions to answer about how endothelium responds to shear stress and other mechanical forces. We would like to know, for example, the first event or events when shear stress increases and the downstream consequences as the coordinated system response unfolds. We would like to know how these processes vary depending on the type of blood vessel or lymphatic, the context, the direction and chaos of the shear stress and the co-applied forces such as membrane tension and radial stretch. We are a long way from fully understanding. Instead, we know some molecules that are critical and a plethora of molecular changes that occur over time. What we lack is a time-series of events and their spatial orchestration. A key idea emerging from our work and that of others is that PIEZO1 channels are extremely rapid and sensitive responders to mechanical force sitting at the heart of this biology. Our hypothesis is that activation of these channels is a first or near-first event when force is applied at the endothelial surface or glycocalyx. This puts PECAM1 downstream. Consistent with this model, tyrosine phosphorylation of PECAM1 measured 5 minutes after the start of shear stress depends on PIEZO1 (Ref. 32). However, we need much more sophisticated experiments to understand what happens. Our findings suggest an approach in which it would be necessary to measure in real-time the protein structural conformations of PIEZO1 and PECAM1 at apical and junctional membranes during shear stress. This would ideally be complemented by experiments incorporating specific disruption of the PIEZO1-PECAM1 interaction, thereby enabling testing of its role and the role of the proposed negative feedback of PECAM1 on PIEZO1.”

2. Another interesting observation is that an increase in CD31 expression correlates with and

might be causative for reduced sensitivity of Piezo1 to pressure, suggesting a negative feedback but again it is not clear what is the role of this negative feedback in the shear stress-induced mechanosensation.

We tried to clarify and contextualize this result better:

Lines 378-380: “We suggest negative feedback from PECAM1 to PIEZO1 when junctional density becomes high, serving to dampen PIEZO1’s remodelling role.”

Lines 781-783: “Negative feedback is suggested to occur from PECAM1 to PIEZO1 as junctional intensity increases to enable PIEZO1’s role in driving junctional remodelling to be suppressed once remodelling is complete and junctions need to return to a tighter, less leaky, state.”

We suggest that the negative feedback is a feature of the junctional PIEZO1. The apical PIEZO1 may be relatively free of this because CD31 (PECAM1) is junctional, enabling apical PIEZO1 to respond better to shear stress.

The critical question is whether CD31 is required for piezo1-induced/dependent downstream flow responses.

We worked to address this matter through the discussion indicated above. CD31 is a key protein in endothelial cells, so its depletion has wide-ranging effects that could be beyond disabling of a simple chain of events. There are likely to be structural changes as cell-cell junctions alter. We therefore think we should be cautious about interpreting effects of CD31 knockdown and that we can only determine the role of CD31 in PIEZO1 signalling by first delineating specific aspects of its interaction with PIEZO1.

3. The data about the effects of Piezo1 on the structural integrity of CD31 and VE-cadherin junctions show that there is an effect on the width of the junctions upon depletion/re-introduction of extracellular Ca in the presence vs. the absence of Piezo1. More specifically, they show that Piezo1 KD results in thinner junctions. The structural effect is evident but the functional implications are entirely not clear. In many studies, an increase in junctional width is considered a sign of leakiness. It needs to be tested if this is the case here and if yes, what does it mean for the mechanotransduction.

Thank you. We corrected and improved our text and associated figures, leading to a clearer concept (especially seen in the new Figures 7 and 8 and associated text).

We did not specifically study leakiness because there is extensive work published on this (**Lines 96-110**). We made many changes, however, to address this point, noting in particular:

Lines 324-340: “**Physiological PIEZO1 increases junction width and radial actin in endothelial cells** CDH5 is Ca²⁺ regulated and so PIEZO1 channels could serve to regulate Ca²⁺ locally and thereby link local mechanical force to adherens junction structure. Consistent with this hypothesis, force-induced junctional remodelling in HUVECs is associated with transient elevation of cytosolic Ca²⁺ concentration sufficient for junctional remodelling (Ref. 46). We depleted and then reintroduced extracellular Ca²⁺ to observe Ca²⁺ regulated junction formation in confluent HUVECs (Figure 7A), testing the role of PIEZO1 by depleting it (but not PECAM1 or CDH5) using PIEZO1 targeted siRNA (Figure S19). In the PIEZO1-depleted group, there are thinner (tighter) junctions in the +calcium (post) condition (Figure 7A-E, Figure S20). We also stained F-actin using phalloidin 568 (Figure S21A-C) because cytoskeletal architecture is coordinated with junctional remodelling and affected by stretch (Ref. 46). In the PIEZO1-depleted condition, there are fewer F-actin peaks in cross-section, suggesting less radial actin (which spans focal adhesions) and more cortical actin (which coordinates with cell

junctions). This is most striking when extracellular Ca^{2+} is returned after Ca^{2+} depletion (+calcium (post) in Figure S21D-F). The data suggest that PIEZO1 increases the width of junctions, consistent with the junctions being less tight and more able to remodel. Moreover, PIEZO1 promotes radial actin, which is also consistent with PIEZO1 facilitating cell and junctional remodelling (Figure 7F, G)."

Lines 446-455: "The endothelial response to shear stress may often occur alongside adaptive changes in endothelium such as cytoskeletal rearrangements and junction remodelling in diapedesis, endothelial remodelling, inflammation or other events. PIEZO1 is important here too, apparently existing as an adaptable mechanical force-sensing cassette in multiple subcellular compartments. PIEZO1 stimulation straightens CDH5 junctions and its depletion inhibits stretch-evoked remodelling (Refs. 45, 46). PIEZO1 also has roles in focal adhesions (Refs. 30, 77), to which radial actin is attached, consistent with a remodelling role of PIEZO1. Although our data and that of other investigators support roles for PIEZO1 as an enabler of junctional remodelling, whether this results in tighter or weaker junctions (less or more permeability) may depend on context as PIEZO1 depletion strikingly enhances or suppresses vascular permeability in vivo (Refs. 42, 43)."

4. Similarly, a lack of Piezo1 is shown to result in thinner junctional actin, which again is not clear whether it indicates a stronger or weaker cortical actin. An increase in cortical actin is expected to strengthen the junctions but an increased width might indicate weakening of junctions. This needs to be clarified.

Thank you. We revised this section. Please see our response above to Point 3.

Also, similarly to the previous comments, a connection with mechano-sensitivity is only implied, not shown

We show flow data for molecular interaction of PIEZO1 with CD31 triad (e.g. new Figure 6). Still, after 12 years of research on PIEZOs, the only known physiological stimulator of PIEZO1 is mechanical force, so we imply mechano-sensitivity based on this and emphasized the point in the revised manuscript (e.g., **Lines 371-373**): "we note the compelling evidence that PIEZO1 channels are direct and specific sensors of mechanical force, apparently having evolved for this purpose (Refs. 25-27), including in endothelial cells (Refs. 30, 38, 39, 63)."

Reviewers' comments:

Reviewer #1 (Remarks to the Author):

MS COMMSBIO-22-1039-T, Chuntharpursat et al. Piezo1 channel and CD31 (PECAM1) partnership in endothelial force sensing

Shear stress exerted on a blood vessel wall is a physical phenomenon resulting from the frictional forces generated by the blood flow on the luminal side of the endothelial wall. In contrast to the pressure impulse, which causes a radial stretching of e.g. the arterial wall, the frictional forces of the blood generate a tangential shear stress to the blood flow direction and to the vessel axis, which depends on the instantaneous viscosity of the fluid. It is significantly lower than other mechanical stresses, which means that the sensitivity to shear stress is in a completely different range than other types of mechanosensitivity.

Recent studies have shown that the mechanoregulated plasma membrane ion channel Piezo1 and the mechanocomplex of PECAM1 (CD31), vascular endothelial cadherin (VE-cadherin) and VEGFR2/3 complex, located at cell-cell junctions, are important molecular players in the sensitivity of the endothelium to fluid shear stress. However, the nature of their interactions and whether Piezo1 and/or PECAM1 act as a primary stress sensor have remained controversial until now.

This manuscript intends to describe the localization of Piezo1 at adherens junction, interactions with CD31/PECAM1 and VE-cadherin in the context of cell-cell junction formation, and - in line with previous reports (Nourse et al. 2017, *Semin. Cell Dev. Biol.* 71, 3-12) - the relationship to the EC cytoskeleton. Of particular interest is the authors' hypothesis that CD31 and VE-cadherin are indirectly regulated via Piezo1 by adjusting local calcium levels that influence the remodelling of cell-cell contacts in epithelia. The results are based on optical co-localization studies in mouse endothelial cells, cell-cell contacts of different model cell systems expressing CD31, VE-cadherin and Piezo1, and biochemical experiments. The authors reach their conclusion by combining a variety of different experimental approaches in which Piezo1 is activated either by fluid stress or chemically (Yoda1). However, the manuscript seems to lack a clear experimental rationale and pooled too much data, with often inadequate data analysis and statistics. The introduction does not clearly describe the experimental focus and seems to abruptly move from a general description to a very specific molecular description. I recommend that the manuscript be thoroughly revised, probably involving a selection of key experiments, and clearly indicating the biophysical analysis (see specific comments below).

Specific comments:

1. Line 116 and following: In Figures 1A-C and S1E-G, the authors show patch-clamp recordings of endothelium cell membranes and note that the Piezo1HA channels respond similarly to wild-type channels in the presence of fluid shear stress. This is not made clear enough, in my opinion, as completely different figures and original data are shown for wild-type and mutant channels. What is the origin of the noisy current signal from Piezo1HA in Figure 1A? Why were the data points in Figure 1F connected to each other? A clear side-by-side comparison of the data from both channels would be helpful. In order to compare the conductivities of Piezo1 with the literature, it would also be helpful if the electrolyte concentrations were given in the figure legend. In this context, I also wonder whether the specification and statistics of conductivities to the first decimal is justified for relatively small group numbers of $n < 10$. More clarity is essential, as the first figure sets the stage for the rest of the manuscript.

The authors show that gadolinium blocks the activity of mechanosensitive channels in isolated EC membranes. However, since gadolinium can block the activity of several cation-specific MSCs, it seems important to show that more specific chemical compounds can activate or inhibit Piezo1

channel activity. Another control could be to determine whether Piezo1 acts through its mechanosensitive property and triggers ion channel activities when the cell membrane is stretched by applying negative pressure via the micropipette attached to the membrane.

2. Line 128 - : The authors state that Piezo1 is expressed in retinal venous and microvascular endothelia. Why is a HA intensity signal measured in - CD31 and +CD31 wild-type ECs when the mice do not express HA protein? In my opinion, the dHA signals of about 0.4 afu in figures 1 K,L, Q and R are merely a background. It is also not clear to me what the authors are trying to say with figures 1 I, J, O and P, especially since the HA signals in all tracks between 0 and 20 afu are similar. Why do the HA intensity values in figures I, J, O and P not match those in figures K, L, Q and R?

3. Line 143 - : Co-localization image analyses aim to find indications of possible protein-protein interactions. The latter can be determined by superimposing two images and examining the appearance of the combined color. However, this intuitive method can only work if the intensity values of the two images are similar. The latter does not seem to be the case in Figures 1 I, j, O and P. Based on the information in Materials and Methods (lines 952-974), I am not entirely convinced how to conclude from figures 1G and 1S, U and V that Piezo1 co-localizes with CD31 in the endothelium of retinal vein, unless co-localization is assessed in more quantitative and statistical detail.

4. Line 148 - : Figures 2 A-D show Piezo1 and CD31 in Cos7 cells after heterologous expression by STED microscopy. I wonder how individual Piezo1 and CD31 molecules were identified and why this was done "manually" in selected regions of interest. I have the impression that both proteins are also present in non-cell contact regions. What is meant by selecting particles with a circularity between 0 and 1? Doesn't that mean that any signal with any structure was selected? In general, there are two categories of quantitative approaches to co-localization analysis: intensity-based methods, which calculate global measures using the correlation information of the intensities of two channels ("colors"), and object-based methods, which first segment and identify objects and then consider object distances. Better statistics ($n > 3$) and a clearer step-by-step representation of how the center of mass and the distances between Piezo1 and CD31 were determined in both functional and non-functional regions would strengthen the conclusions drawn from the data. Figure S2A indicates accuracies of distances in the range of 1 nm. How were they determined at a pixel size of 20 nm? What do the authors mean by "parameter uncertainties" in line 172? Since there is no punctate staining of CD31 in Fig. 2B, suggesting that CD31 is very abundant at cell-cell contact sites, I wonder how the histograms shown in Fig. 2E and S2B change when CD31 and/or Piezo1 expression is reduced and when CD31 is mutated at the N- and C-terminus, respectively, to prevent proximity to Piezo1. If the expression level of CD31 or Piezo1 is high, shouldn't the distances between CD31 and Piezo1 automatically be low? In this context, it would be interesting to compare the closest and second closest distances between CD31 and Piezo1 with those of the individual proteins themselves. Another important question is how the effective resolution of the STED micrographs was determined, which is particularly relevant to judge the correctness of Fig. 2E.

5. Line 187 - : While lifetime measurements are suitable to measure the proximity between donor and acceptor dyes, accurate distance measurements are very dependent on the underlying assumptions, especially orientation, motional and concentration effects. It is unclear how the authors approached this question (Fig. 2F-Q).

6. Line 237 - : The authors conclude from patch-clamp experiments that CD31 reduces the mechanical sensitivity of Piezo1 channels when expressed in modified HEK293 cells. Since the error bars are significantly overlapping and to further support this conclusion, it is essential to significantly improve the statistics of the data representing the activation range and to compare the stimulus-response curves with the literature.

7. Line 322 - : The conclusions from Fig. 5, in particular the distance between junctions, do not seem to be justified by the data. In panels Fig. 5 D & E and I & J, the median values appear to be extremely

close. It was also not clear how the intersections were selected for the analyses; what does "manually using a polygon" (line 879) mean?

Minor points:

1. The conclusions drawn from Fig. 1 should be formulated more cautiously.
2. Buffer concentrations should be reported in the same way throughout the manuscript (e.g. "... contains (mM) 135 KCl, 4 KCl ... (lines 923-933)" as opposed to "... 0.1 mM CaCl₂, 0.1 mM MgCl₂ ... (lines 952 - 955)").
3. The legend to figure 2F-H ("FRET/FLIM studies") could be more detailed.

Reviewer #2 (Remarks to the Author):

This manuscript addresses an important topic in the field of vascular mechanobiology: how are linked two of the major mechanosensors, the VE-cadherin junctional complex comprising PECAM, VE-cadherins and VEGFRs on one side and Piezo1 on the other side? The authors suggest there might be a direct link, through a close association of both Piezo1 and PECAM at the endothelial cells junctions.

They engineered a unique mouse strain which expresses a small tag inside Piezo1 sequence, to allow for easy detection and capture from tissues. The function of Piezo1 is not affected by the HA tag, as described by functional patch clamp experiments.

Starting from there, the authors investigate Piezo1-HA localization in blood vessels from the retina, which has the advantage to display three different types of vessels: veinules, arterioles and capillaries. Their first observation, of interest, is that Piezo1-HA is not expressed uniformly. It is also confirmed by a functional experiment whereas Yoda1, a Piezo1 agonist triggers the disassembly of PECAM at the cell-cell junctions.

Recommendation 1: please show complete image of the mouse retina with anti-HA staining.

Recommendation 2: image seems dirty, can you improve the staining, maybe with a different blocking protocol? It is too late now that the study is done but other tags might have improved resolution (Spot-tag for example). The objectives used for the retinas are not specified, as well as the resolution of the images. It would have been nice to get higher resolution images, the use of the term "colocalization" with those images is not adequate. you can only state that Piezo1-HA appears to localise at the junctions.

Recommendation 3: can you also image bigger vessels such as veins and large arteries (aorta for example)?

Recommendation 4: Effect of Yoda1 might be indirect (huge calcium entry which triggers changes in cell-cell junction structure), it should therefore be explicit to the reader.

Recommendation 5: PECAM should be used instead of CD31.

The authors pursue their study by using COS7 cells expressing both PECAM and Piezo1. They make the argument that those cells are adequate because they do not normally express both proteins. I appreciate the strategy but Piezo CRE recombinase KO mice are available, as well as PECAM KO mice and could have provided endothelial cells without endogenous piezo or PECAM. Indeed, cell-cell junctions in endothelial cells are vastly different from the ones observed in monolayers of COS7 (no

VE-Cad nor VEGFRs are present for example). the other issue is that over expressed PECAM and Piezo1, which is a massive protein complex, will most likely meet and colocalize at the plasma membrane and cell-cell junctions. the image displayed in figure 2A in fact argues against a specific localization: only a tiny portion of the whole cell-cell junction(highlighted) seems to show some specific accumulation of both proteins, most of the time, they seem to be randomly expressed all over the plasma membrane. I am therefore not a big fan of this data set and findings should be toned down or reproduced in endothelial cells.

The authors pursue by performing investigation of direct interaction between the two proteins.

Recommendation 1: Figure 3A is not acceptable in its current form, please provide full blots and images.

Recommendation 2: HEK293 cells are not the best cells to use to perform this experiment. I would recommend to use confluent endothelial cells.

Then they move on a provocative, yet fascinating, idea: PECAM association with Piezo1 would impairs piezo activity. It is well known that Piezo1 quickly inactivates and is therefore almost never fully opened, which makes a lot of sense from a physiological perspective. I like the proposed idea and the data support it. however, I still question the relevance of the model used. The involvement of the N terminal part is adequately supported by a co-IP experiment, although the results are not black or white.

Figure 3I blot is not acceptable in its current form.

Then the authors investigate the role of Piezo1 in the formation of the cell-cell junctions and I have one major comment here: if Piezo1 was necessary, how can you explain that arteries have beautiful cell-cell adherent junctions as you show in figure 1, while they are also lacking Piezo1?

All in all, the authors have identified a very important question and generated intriguing data but this manuscript fails to capture the subject in its whole complexity because the authors tried to do too much, with, in my opinion, inadequate methodology and by jumping too quickly to conclusions. I would recommend them to only focus on some parts of the manuscript, to tone down their claims and to improve a bit the models and data from the mice. There are definitely some aspects of the study which are potentially ground breaking but not in its current form.

Reviewer #3 (Remarks to the Author):

The study addresses a very important goal of finding the connection between two major mechanosensitive endothelial pathways, regulated by PECAM 1 junctional protein and Piezo1 ion channel respectively in order to reconcile the competing hypotheses for endothelial mechanosensation. The study presents multiple lines of evidence using most state-of-the-art approaches to demonstrate physical and functional interactions between the two pathways. The authors also present interesting mechanistic insights into the structural requirements of the interaction. However, the overall picture of how the two pathways interact to confer endothelial shear stress sensitivity remains vague and not clear. There is also no direct evidence that activation of piezo1 by mechanical forces is transduced to the activation of CD31/VE-cadherin junctional complex.

Specific Concerns:

1. The authors show that Piezo1 co-localizes with CD31 in areas of cell-cell contacts, the interaction is verified by FRET. It is also shown that the activation of Piezo1 with its known activator Yoda disrupts CD31 endothelial junctions, but the dispersion of CD31 upon activation of Piezo1 seems to contradict the known effects of laminar flow on the junctions. The more relevant question would be whether Piezo1 activity initiates or enables flow-induced CD31 response.
2. Another interesting observation is that an increase in CD31 expression correlates with and might be causative for reduced sensitivity of Piezo1 to pressure, suggesting a negative feedback but again it is not clear what is the role of this negative feedback in the shear stress-induced mechanosensation. The critical question is whether CD31 is required for piezo1-induced/dependent downstream flow responses.
3. The data about the effects of Piezo1 on the structural integrity of CD31 and VE-cadherin junctions show that there is an effect on the width of the junctions upon depletion/re-introduction of extracellular Ca in the presence vs. the absence of Piezo1. More specifically, they show that Piezo1 KD results in thinner junctions. The structural effect is evident but the functional implications are entirely not clear. In many studies, an increase in junctional width is considered a sign of leakiness. It needs to be tested if this is the case here and if yes, what does it mean for the mechanotransduction.
4. Similarly, a lack of Piezo1 is shown to result in thinner junctional actin, which again is not clear whether it indicates a stronger or weaker cortical actin. An increase in cortical actin is expected to strengthen the junctions but an increased width might indicate weakening of junctions. This needs to be clarified. Also, similarly to the previous comments, a connection with mechano-sensitivity is only implied, not shown

Manuscript COMMSBIO-22-1039A

Reviewer comments are in blue.
Author responses are in black.

Response to Reviewer #1

Reviewer #1 (Remarks to the Author):

Manuscript COMMSBIO-22-1039-T

The experimental rationale and focus of the manuscript have been improved by a complete restructuring of figures, tables and text.

Thank you.

Nevertheless, still some important questions must be answered more carefully. Since I don't think any new data has been added and the data is still based on relatively small populations, I wonder if it is sufficiently convincing statistically.

We answer the questions below. We provide new data, data analysis, substantial revision throughout and full transparency for all the data and unbiased statistical analysis.

Regarding my initial comment on Fig. 1, I welcome that the authors have now improved this figure to show a direct side-by-side comparison of patch-clamp data from wild-type and mutant PIEZO1 channels. I also welcome that the authors have now included the electrolyte concentrations in the figure legend. The data suggest that both more PIEZO1 wild-type and mutant channels switch between open and closed in the presence of shear flow. If I were to try to do this experiment, is this a representative observation (1 open channel in the absence of flow vs. 3 channels in the presence of flow and a membrane potential) or does it depend on the patch selected to be studied?

The cell membrane area is randomly selected for patching. The observed ionic currents are a bit different for each patch, but the currents are always from these 25 pS channels. We found the number of simultaneous openings in each patch to be between 1 and 3 for HA and WT channels depending on flow, as you rightly suggest and as we display through our the selection of representative traces (Figure 1A, F). Data from all patches studied are included without selection in the analysis (NP_o data of Figure 1K; $n=6-7$ per group).

PIEZO1 has been shown to rectify outward at more positive voltages in symmetric electrolyte solution (Moroni et al. 2018 Nat Comm 9 1096). The inward conductance was significantly higher than the outward conductance (41 pS vs. 27 pS). Is there a reason why the I/V curves are only displayed for asymmetrical voltages from -75 mV to +30 mV? I see no reason why the I/V curves are not also displayed up to higher voltages.

The physiological membrane potential for endothelial cells is in the negative voltage range (Figure 1D, E, I, J). We spanned -75 to +30 mV to give broader insight because the positive voltage data might be of interest to investigators studying other cell types, which may experience such voltages during action potential firing.

Fig. 2B shows a cross-section of the PIEZO1-HA and PECAM1 signals, which should show the colocalization of PECAM 1 and PIEZO1. Although I am aware that image analysis of real cells, such as those taken from the retinal vein, is complicated, I am still not convinced by Fig. 2B (as well as Figs. 5G and 5H; Figs. S4B, S4F and S4G). Do you see the same overlap of

HA and PECAM1 signals (greyed out) when selecting the cross-section at a different location? This could be improved.

We inserted in Figure S4 (D, E) data for a different location for comparison with Figure 2B. The result is similar to that shown in Figure 2B and so we think Figure 2B is representative. Mean data from all images/experiments of this type are in Figure 2C.

In Fig. 8B, the authors have attempted to validate their results by copurifying PIEZO1 and PECAM1. However, I do not see how it can be concluded beyond doubt that PIEZO1 and PECAM1 "are associated after size exclusion chromatography" (line 197).

We wrote "and then purified by size-exclusion chromatography", not "are associated after size exclusion chromatography".

The data show detection of both proteins after purification and size exclusion chromatography (Figure S8B).

After SDS-PAGE, PIEZO1 was stained with Coomassie, but PECAM1 was only detected with a PCAM1-specific antibody. Doesn't this suggest that PIEZO1/PECAM1 rather form a small non-stoichiometric complex population? Why was PECAM1 not stained with Coomassie or at least silver to exclude other proteins that might mediate the interaction between PIEZO1 and PECAM1?

The need for antibody detection indicates a relatively small amount and this is consistent with our conclusion of two pools of PIEZO1 – one associated with and one not associated with PECAM1. We suggest that the associated pool is at cell-cell junctions, which are small but important. We investigated this pool in situ using FRET/FLIM.

When characterizing protein complexes by size exclusion chromatography, it is necessary to visualize all relevant protein fractions eluted from the column and look for overlapping fractions with a similar relative ratio of PIEZO1 and PECAM1.

We included all eluted proteins. We tried to match the protein expressions but recognise that this is technically challenging for proteins of different size and properties.

It is important to check whether PIEZO1 and PECAM1 are also found in other fractions to really check whether the two proteins are stably associated. PIEZO1 and PECAM1 should, in my opinion, be colocalized in the same fractions.

We isolated cell membranes but did not fractionate them. While supporting our hypothesis, the HEK 293 cell data have limited value here because cell-cell junctions do not form clearly, reducing the value of membrane fractionation. To investigate colocalization where it occurs (i.e., at junctions), we used the in-cell/in situ technical approach of FRET/FLIM in COS-7 cells. We discuss the value of this approach in the manuscript.

We revised and added to the Discussion: "We successfully copurified PIEZO1 and PECAM1 overexpressed in HEK 293 cells, consistent with there being interaction, but this should be seen in the context of the central idea that there are two pools of PIEZO1 – one associated with and one not associated with PECAM1. We suggest that the associated pool is at cell-cell junctions, which are small but important. We investigated this pool in situ using FRET/FLIM and based on the arising data suggest stable association at cell-cell junctions of quiescent cell monolayers after junctions have formed. While the results of the HEK 293 cell experiments support our hypothesis, the value of this approach is limited in the context of such complex biology. Investigation at the site where the association occurs is especially important."

What was the molecular mass of the PIEZO1/PECAM1 complex determined by size exclusion chromatography?

We can expect it to be large and difficult to determine. Each PIEZO1 is about 300 kDa with glycosylation (Figure S8A, B and D), so each PIEZO1 channel should be about 900 kDa. Each PECAM1 is about 130 kDa with glycosylation (Figures 3D and S14B). The stoichiometry is unknown but 1:1 stoichiometry would suggest a total mass of about 1,300 kDa. We used a Vivaspin centrifugal filter (300 kDa MWCO) to concentrate the complex before running the proteins on a gel, where they separated (Figure S8B).

Since the interaction between the two proteins does not seem to be very strong, the reader occasionally has the feeling that a variety of biochemical and optical approaches are used to prove their interaction and the formation of a complex, since each method in itself does not provide clear and absolutely unambiguous results.

Our data suggest association involving one pool of PIEZO1 channels at a specific subcellular localization (cell-cell junctions). Several approaches gave independent perspectives on this challenging biology, which is a strength of the study.

A careful gel filtration analysis would be convincing (1 profile for each single protein, 1 profile for the putative complex, all fractions).

We address this above.

Manuscript COMMSBIO-22-1039A

Reviewer comments are in blue.
Author responses are in black.

Response to Reviewer #3

Reviewer #3 (Remarks to the Author):

The authors provided thoughtful responses, added some new data and significantly improved the discussion of the manuscript. This is now acceptable for publication

Thank you.